# Differentially Private Empirical Risk Minimization under the Fairness Lens

**Cuong Tran**
Syracuse University
ctran@syr.edu

**My H. Dinh**
Syracuse University
mydinh@syr.edu

**Ferdinando Fioretto**
Syracuse University
ffiorett@syr.edu

## Abstract

Differential Privacy (DP) [14] is an important privacy-enhancing technology for private machine learning systems. It allows to measure and bound the risk associated with an individual participation in a computation. However, it was recently observed that DP learning systems may exacerbate bias and unfairness for different groups of individuals [3, 27, 23]. This paper builds on these important observations and sheds light on the causes of the disparate impacts arising in the problem of differentially private empirical risk minimization. It focuses on the accuracy disparity arising among groups of individuals in two well-studied DP learning methods: output perturbation [11] and differentially private stochastic gradient descent [2]. The paper analyzes which data and model properties are responsible for the disproportionate impacts, why these aspects are affecting different groups disproportionately, and proposes guidelines to mitigate these effects. The proposed approach is evaluated on several datasets and settings.

## 1 Introduction

While learning systems have become instrumental for many decisions and policy operations involving individuals, the use of rich datasets combined with the adoption of black-box algorithms has sparked concerns about how these systems operate. Two key concerns regard how these systems handle discrimination and how much information they leak about the individuals whose data is used as input.

Differential Privacy (DP) [14] has become the paradigm of choice for protecting data privacy and its deployments are growing at a fast rate. DP is appealing as it bounds the risks of disclosing sensitive information of individuals participating in a computation. However, it was recently observed that DP systems may induce biased and unfair outcomes for different groups of individuals [3, 23, 27]. The resulting outcomes can have significant societal and economic impacts on the involved individuals: classification errors may penalize some groups over others in important determinations including criminal assessment, landing, and hiring [3] or can result in disparities regarding the allocation of critical funds, benefits, and therapeutics [23]. *While these surprising observations have become apparent in several contexts, their causes are largely understudied and not fully understood.*

This paper makes a step toward addressing this important knowledge gap. It builds on these key observations and sheds light on the causes of the disparate impacts arising in the problem of differentially private empirical risk minimization (ERM). It focuses on the accuracy disparity arising among groups of individuals in two well-studied DP learning methods: output perturbation [11] and differentially private stochastic gradient descent (DP-SGD) [2]. The paper analyzes which properties of the model and the data are responsible for the disproportionate impacts, why these aspects are affecting different groups disproportionately, and proposes guidelines to mitigate these effects.

In summary, the paper makes the following contributions:
1. It develops a notion of fairness under private training that relies on the concept of excessive risk.

2. It analyzes this fairness notion in two DP learning methods: output perturbation and DP-SGD.

3. It isolates the relevant components related with noise addition and gradient clipping responsible for the disparate impacts.

4. It studies the behaviors and the causes for these components to affect different groups of individuals disproportionately during private training.

5. Based on these observations, it proposes a mitigation solution and evaluates its effectiveness on several standard datasets.

To the best of the authors knowledge, this work represents a first step toward a deeper understanding of the causes of the unfairness impacts in differentially private learning.

## 2 Related work

The research at the interface between differential privacy and fairness is receiving increasing attention and can be broadly categorized into three main lines of work. The first shows that DP is in alignment with fairness. Notable contribution in this direction include Dwork et al. [15] seminal work, which highlights the relation between individual fairness and differential privacy, and Khalili et al. [19], which shows that the private exponential mechanism can produce fair outcomes in some selection problems. Works in the second category study the setting under which a fair model can leak privacy [22, 18, 9, 25, 28]. These works propose learning frameworks that guarantee DP while also encouraging the satisfaction of different notions of fairness. For example, Xu et al. [27] proposes a private and fair variant of DP-SGD that uses separate clipping bounds for each groups of individuals. Such proposal encourages accuracy parity at the expense of an extra privacy cost (required to customize the clipping bound for each group). Works in the last category show that private mechanisms can have a negative impact towards fairness [23, 27, 3, 17, 25]. For example, Cummings et al. [13] shows that it is impossible to achieve *exact* equalized odds while also satisfying pure DP. Pujol et al. [23] observe that decisions made using a private version of a dataset may disproportionately affect some groups over others. Similar observations were also made in the context of model learning. Bagdasaryan et al. [3] empirically observed that the accuracy of a DP model trained using DP-SGD drops disproportionately across groups causing larger negative impacts to the underrepresented groups. Farrand et al. [17] reaches similar conclusions. The authors empirically show that the disparate impact of differential privacy on model accuracy is not limited to highly imbalanced data and can occur even in situations where the classes are slightly imbalanced.

This paper builds on this body of work and their important empirical observations. It derives the conditions and studies the causes of unfairness in the context of private empirical risk minimization problems as well as it introduces mitigating guidelines.

## 3 Preliminaries

Differential privacy (DP) [14] is a strong privacy notion used to quantify and bound the privacy loss of an individual participation to a computation. Informally, it states that the probability of any output does not change much when a record is added or removed from a dataset, limiting the amount of information that the output reveals about any individual. The action of adding or removing a record from a dataset $D$, resulting in a new dataset $D'$, defines the notion of *adjacency*, denoted $D \sim D'$.

**Definition 1.** *A mechanism $\mathcal{M}: \mathcal{D} \to \mathcal{R}$ with domain $\mathcal{D}$ and range $\mathcal{R}$ is $(\epsilon, \delta)$-differentially private, if, for any two adjacent inputs $D \sim D' \in \mathcal{D}$, and any subset of output responses $R \subseteq \mathcal{R}$:*

$$\Pr[\mathcal{M}(D) \in R] \le e^{\epsilon} \Pr[\mathcal{M}(D') \in R] + \delta.$$

Parameter $\epsilon > 0$ describes the *privacy loss* of the algorithm, with values close to 0 denoting strong privacy, while parameter $\delta \in [0, 1)$ captures the probability of failure of the algorithm to satisfy $\epsilon$-DP. The global sensitivity $\Delta_\ell$ of a real-valued function $\ell : \mathcal{D} \to \mathbb{R}^k$ is defined as the maximum amount by which $\ell$ changes in two adjacent inputs: $\Delta_\ell = \max_{D \sim D'} \|\ell(D) - \ell(D')\|$. In particular, the Gaussian mechanism, defined by $\mathcal{M}(D) = \ell(D) + \mathcal{N}(0, \Delta_\ell^2 \sigma^2)$, where $\mathcal{N}(0, \Delta_\ell^2 \sigma^2)$ is the Gaussian distribution with 0 mean and standard deviation $\Delta_\ell^2 \sigma^2$, satisfies $(\epsilon, \delta)$-DP for $\delta > \frac{4}{5} \exp(-(\sigma\epsilon)^2/2)$ and $\epsilon < 1$ [16].

## 4    Problem settings and goals

The paper adopts boldface symbols to describe vectors (lowercase) and matrices (uppercase). Italic symbols are used to denote scalars (lowercase) and data features or random variables (uppercase). Notation $\|\cdot\|$ is used to denote the $L_2$ norm. The paper considers datasets $D$ consisting of $n$ individuals' data points $(X_i, A_i, Y_i)$, with $i \in [n]$ drawn i.i.d. from an unknown distribution. Therein, $X_i \in \mathcal{X}$ is a feature vector, $A_i \in \mathcal{A}$ is a protected group attribute, and $Y_i \in \mathcal{Y}$ is a label. For example, consider the case of a classifier that needs to predict the risks associated with a lending decision. The training example features $X_i$ may describe the individual's demographics, education, credit score, and loan amount, the protected attribute $A_i$ may describe the individual gender or ethnicity, and $Y_i$ represents whether or not the individual will default on the loan. The goal is to learn a classifier $f_{\boldsymbol{\theta}} : \mathcal{X} \to \mathcal{Y}$, where $\boldsymbol{\theta}$ is a vector of real-valued parameters, that guarantees the *privacy* of each individual data $(X_i, A_i, Y_i)$ in $D$. The model quality is measured in terms of a nonnegative *loss function* $\ell : \mathcal{Y} \times \mathcal{Y} \to \mathbb{R}_+$, and the problem is that of minimizing the empirical risk (ERM) function:

$$\min_{\boldsymbol{\theta}} \mathcal{L}(\boldsymbol{\theta}; D) = \frac{1}{n} \sum_{i=1}^{n} \ell(f_{\boldsymbol{\theta}}(X_i), Y_i). \tag{L}$$

For a group $a \in \mathcal{A}$, the paper uses $D_a$ to denote the subset of $D$ containing exclusively samples whose group attribute $A = a$. The paper focuses on learning classifiers that protect the disclosure of the individuals' data using the notion of differential privacy and it analyzes the fairness impact (as defined next) of privacy on different groups of individuals. Importantly, the paper assumes that the attribute $A$ is not part of the model input during inference.

**Fairness** The fairness analysis focuses on the notion of *excessive risk*, a widely adopted metric in private learning [26, 29]. It defines the difference between the private and non private risk functions:

$$R(\boldsymbol{\theta}, D) = \mathbb{E}_{\tilde{\boldsymbol{\theta}}} \left[ \mathcal{L}(\tilde{\boldsymbol{\theta}}; D) \right] - \mathcal{L}(\boldsymbol{\theta}^*; D), \tag{1}$$

where the expectation is defined over the randomness of the private mechanism and $\tilde{\boldsymbol{\theta}}$ denotes the private model parameters while $\boldsymbol{\theta}^* = \operatorname{argmin}_{\boldsymbol{\theta}} \mathcal{L}(\boldsymbol{\theta}; D)$. The paper uses shorthands $R(\boldsymbol{\theta})$ and $R_a(\boldsymbol{\theta})$ to denote, respectively, the population-level $R(\boldsymbol{\theta}, D)$ excessive risk and the group level $R(\boldsymbol{\theta}, D_a)$ excessive risk for group $a$. Fairness is measured with respect to the *excessive risk gap*:

$$\xi_a = |R_a(\boldsymbol{\theta}) - R(\boldsymbol{\theta})|. \tag{2}$$

(Pure) fairness is achieved when $\xi_a = 0$ for all groups $a \in \mathcal{A}$ and, thus, a private and fair classifier aims at minimizing the maximum excessive risk gap among all groups. The paper assumes that the private mechanisms are non-trivial, i.e., they minimize the population-level excessive risk $R(\boldsymbol{\theta})$.

All proofs are reported in the Appendix, Section A.

## 5    Warm up: output perturbation

The paper starts with analyzing fairness under the DP setting induced by an output perturbation mechanism. In this setting the analysis restricts to twice differentiable and convex loss functions $\ell$. Output perturbation is a standard DP paradigm in which noise calibrated to the function sensitivity is added directly to the output of the computation. In the context of the *regularized* ERM problem, adding noise drawn from a Gaussian distribution $\mathcal{N}(0, \Delta_\ell^2 \sigma^2)$ to the optimal model parameters $\boldsymbol{\theta}^*$ ensures $(\epsilon, \delta)$-differential privacy [11]. Therein, $\Delta_\ell = 2/n\lambda$ with regularization parameter $\lambda$. The following result sheds light on the unfairness induced by this mechanism.

**Theorem 1.** *Let $\ell$ be a twice differentiable and convex loss function and consider the output perturbation mechanism described above. Then, the excessive risk gap for group $a \in \mathcal{A}$ is approximated by:*

$$\xi_a \approx \frac{1}{2} \Delta_\ell^2 \sigma^2 \left| \operatorname{Tr}(\boldsymbol{H}_\ell^a) - \operatorname{Tr}(\boldsymbol{H}_\ell) \right|, \tag{3}$$

*where $\boldsymbol{H}_\ell^a = \nabla_{\boldsymbol{\theta}^*}^2 \sum_{(X,A,Y) \in D_a} \ell(f_{\boldsymbol{\theta}^*}(X), Y)$ is the Hessian matrix of the loss function, at the optimal parameters vector $\boldsymbol{\theta}^*$, computed using the group data $D_a$, $\boldsymbol{H}_\ell$ is the analogous Hessian computed using the population data $D$, and $\operatorname{Tr}(\cdot)$ denotes the trace of a matrix.*

The approximation above follows form a second order Taylor expansion of the loss function, linearity of expectation, and the properties of Gaussian distributions. It uses that fact that the excessive risk $R_a(\boldsymbol{\theta})$ for a group $a$ can be approximated as $1/2\Delta_\ell^2\sigma^2\,\mathrm{Tr}(\boldsymbol{H}_\ell^a)$. The proof is reported in Appendix A.

Theorem 1 sheds light on the relation between fairness and the difference in the local curvatures of the losses $\ell$ associated with a group and the population and provides a necessary condition to guarantee pure fairness. It suggests that output perturbation mechanisms may introduce unfairness when the local curvatures associated with the loss function of different groups differ substantially from one another. Additionally, the unfairness level is proportional to the amount of noise $\sigma$ or, equivalently, inversely proportional to the privacy parameter $\epsilon$, for a fixed $\delta$. Finally, it also suggests that groups with larger Hessian traces $\mathrm{Tr}(\boldsymbol{H}_a^\ell)$ will have larger excessive risk compared to groups with smaller Hessian traces. An additional analysis on the reasons behind why different groups may have large differences in their associated Hessian traces is provided in Section 8.

Figure 1 illustrates Theorem 1. The plots show the correlation between the excessive risk [1] and the quantity $\mathrm{Tr}(\boldsymbol{H}_z^\ell)$ for each group $z \in \mathcal{A}$, at varying of the privacy loss $\epsilon \in [0.005, 0.5], \delta = 1e^{-5}$ on two datasets. Each data point represents the average of 100 runs of a DP Logistic Regression (obtained with output perturbation) on each group $z \in \mathcal{A}$. Details on dataset and experimental setting are provided in Appendix B and additional experiments in Appendix C. Note the positive correlation between the excessive risk and the

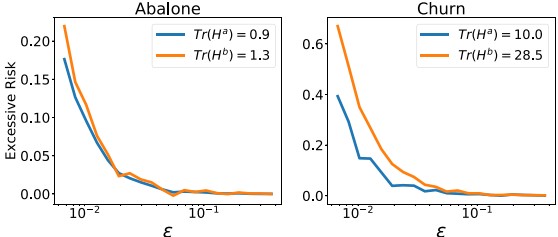

Figure 1: Correlation between excessive risk and Hessian Traces at varying of the privacy loss $\epsilon$.

Hessian trace: *Groups with larger Hessian traces tend to have larger excessive risks*. Note also the inverse correlation between $\epsilon$ and the dependency between the excessive risk and the Hessian trace. This is due to that larger $\epsilon$ values require smaller $\sigma$ values, and thus, as shown in Equation 3, the dependency between the excessive risk and Hessian trace is attenuated.

The following illustrates that even a class of simple linear models may not to satisfy pure fairness.

**Corollary 1.** *Consider the ERM problem for a linear model $f_{\boldsymbol{\theta}}(X) \stackrel{def}{=} \boldsymbol{\theta}^T X$, with $L_2$ loss function i.e., $\ell(f_{\boldsymbol{\theta}}(X), Y) = (f_{\boldsymbol{\theta}}(X) - Y)^2$. Then, output perturbation does not guarantee pure fairness.*

It follows from the observation that the Hessian of the $L_2$ loss for group $a \in \mathcal{A}$, i.e., $\mathrm{Tr}(\boldsymbol{H}_\ell^a) = \mathbb{E}_{X \sim D_a}\,\mathrm{Tr}(XX^T) = \mathbb{E}_{X \sim D_a}\|X\|^2$, depends solely on the input norms of the elements in $D_a$ [2]. Interestingly, this result highlights the relation between fairness and the average input norms of different group elements. When these norms are substantially different one another they will impact their respective excessive risks differently. An additional analysis on this behavior is also discussed in Section 7.

The following is a positive result.

**Corollary 2.** *If for any two groups $a, b \in \mathcal{A}$ their average group norms $\mathbb{E}_{X_a \sim D_a}\|X_a\| = \mathbb{E}_{X \sim D_b}\|X_b\|$ have identical values, then output perturbation with $L_2$ loss function provides pure fairness.*

The above is a direct consequence of Corollary 1. *Note also that pure fairness may be achieved, in this setting, by normalizing the input values for each group independently* (as shown in Appendix C) although this solution requires accessing the sensitive group attributes at inference time.

## 6  Gradient perturbation: DP-SGD

Having identified the dependency between the Hessian of the model loss and the privacy parameters with the excessive risk gap in output perturbation mechanisms, this section extends the analysis to the context of DP Stochastic Gradient Descent (DP-SGD) [2]. In contrast to output perturbation, DP-SGD does not restrict focus on convex loss functions and the privacy analysis does not require optimality of the model parameters $\boldsymbol{\theta}$, rendering it an appealing framework for DP ERM problems.

---

[1]In all experiment presented, the excessive risk is approximated by sampling over 100 repetitions.

[2]Throughout the paper, we abuse notation and treat the dataset $D_Z$ associated with group $Z$ as distributions.

In a nutshell, DP-SGD computes the gradients for each data sample in a random mini-batch $B$, clips their $L_2$-norm, adds noise to ensure privacy, and computes the average. Two key characteristics of DP-SGD are: **(1)** Clipping the gradients whose $L_2$ norm exceeds a given bound $C$, and **(2)** Perturbing the averaged clipped gradients with 0-mean Gaussian noise with variance $\sigma^2 C^2$. The procedure is described in Algorithm 1. Therein, $g_i$ represents the gradient of a data sample $(X_i, A_i, Y_i)$, $\bar{g}_B$ the average

---

**Algorithm 1:** *DP-SGD*

---
**input :** Disjoint dataset $D$ ; Sample prob. $q$; Iterations $T$; Noise variance $\sigma^2$; Clipping bound $C$; learning rate $\eta$
$\theta_0 \leftarrow \mathbf{0}^T$
**for** *iteration $t = 1, 2, \dots T$* **do**
     $B \leftarrow$ random sub-sample of $D$ with $\Pr q$
     **foreach** $(X_i, A_i, Y_i) \in B$ **do**
         $g_i = \nabla \ell (f_{\theta_t}(X_i), Y_i)$
     $\bar{g}_B \leftarrow \frac{1}{|B|} \left( \sum_i \pi_C(g^i) + \mathcal{N}(0, \boldsymbol{I} C^2 \sigma^2) \right)$
     $\theta_{t+1} \leftarrow \theta_t - \eta \bar{g}_B$

---

clipped noisy gradient of the samples in mini-batch $B$, and the function $\pi_C(\boldsymbol{x}) = \boldsymbol{x} \cdot \min(1, \frac{C}{\|\boldsymbol{x}\|})$.

The following theorem is an important result of this section. It connects the expected loss $\mathbb{E}[\mathcal{L}(\boldsymbol{\theta}; D_a)]$ of a group $a \in \mathcal{A}$ with its excessive risk $R_a(\boldsymbol{\theta})$, which is, in turn, used in our fairness analysis. It decomposes the expected loss during private training into three key components: The first relates with the model parameters update and it is not affected by the private training. The other two relate with gradient clipping and noise addition, and, combined, capture the notion of excessive risk.

**Theorem 2.** *Consider the ERM problem* (L) *with loss $\ell$ twice differentiable w.r.t. the model parameters. The expected loss $\mathbb{E}[\mathcal{L}(\boldsymbol{\theta}_{t+1}; D_a)]$ of group $a \in \mathcal{A}$ at iteration $t+1$, is approximated as:*

$$\mathbb{E}\left[\mathcal{L}(\boldsymbol{\theta}_{t+1}; D_a)\right] = \underbrace{\mathcal{L}(\boldsymbol{\theta}_t; D_a) - \eta \langle g_{D_a}, g_D \rangle + \frac{\eta^2}{2} \mathbb{E}\left[g_B^T H_\ell^a g_B\right]}_{\text{non-private term}} \tag{4}$$

$$+ \underbrace{\eta \left( \langle g_{D_a}, g_D \rangle - \langle g_{D_a}, \bar{g}_D \rangle \right) + \frac{\eta^2}{2} \left( \mathbb{E}\left[\bar{g}_B^T H_\ell^a \bar{g}_B\right] - \mathbb{E}\left[g_B^T H_\ell^a g_B\right] \right)}_{\text{private term due to clipping}} \qquad (R_a^{clip})$$

$$+ \underbrace{\frac{\eta^2}{2} \operatorname{Tr}(H_\ell^a) C^2 \sigma^2}_{\text{private term due to noise}} \qquad (R_a^{noise})$$

$$+ O(\|\boldsymbol{\theta}_{t+1} - \boldsymbol{\theta}_t\|^3),$$

*where the expectation is taken over the randomness of the private noise and the mini-batch selection, and the terms $g_Z$ and $\bar{g}_Z$ denote, respectively, the average non-private and private gradients over subset $Z$ of $D$ at iteration $t$ (the iteration number is dropped for ease of notation).*

The result in Theorem 2 follows from a second order Taylor expansion of the non-private and private ERM functions $\mathcal{L}(\boldsymbol{\theta}_t - \eta g_B; D_a)$ and $\mathcal{L}(\boldsymbol{\theta}_t - \eta(\bar{g}_B + \mathcal{N}(0, \boldsymbol{I} C^2 \sigma^2)); D_a)$, respectively, around $\boldsymbol{\theta}_t$ and by comparing their differences. Once again, proofs are reported in Appendix A.

The first term in the expression (Equation (4)) denotes the Taylor approximation of the (non-private) SGD loss. Terms $(R_a^{clip})$ and $(R_a^{noise})$ quantify, together, the excessive risk for group $a$. Therein, $(R_a^{clip})$ quantifies the effect of clipping to the excessive risk, and $(R_a^{noise})$ quantifies the effect of perturbing the average gradients to the excessive risk. Therefore, Theorem 2 shows that there are two main sources of disparate impact in DP-SGD training:

1. *Gradient clipping* $(R_a^{clip})$: which, in turn, depends of three factors: **(i)** The values of the Hessian matrix $H_\ell^a$ of the loss function associated with group $a$; **(ii)** The gradients values $g_{D_a}$ associated with the samples of group $a$; and **(iii)** The clipping bound $C$, which appears in $\bar{g}_B$ and $\bar{g}_D$.
2. *Noise addition* $(R_a^{noise})$: which, in turn, depends on two factors: **(i)** The values of the (trace of the) Hessian matrix $H_\ell^a$ of the loss function associated with group $a$; and **(ii)** The privacy loss parameters $(\epsilon, \delta, \Delta_\ell)$ (which, in turn, are characterized by the noise variance $C^2 \sigma^2$).

A schematic representation of these factors is shown in Figure 2. Therein, $X_{D_a}$ denotes the features values $X \in \mathcal{X}$ of the subset $D_a$ of $D$. *Theorem 2 entails that unfairness occurs whenever different groups have different values for any of the gradient clipping and noise addition excessive risk terms.*

The next sections analyze the reasons behind the disparity in excessive risk focusing, independently, on terms $R_a^{clip}$ (Section 7) and $R_a^{noise}$ (Section 8). Independently studying these terms is motivated by

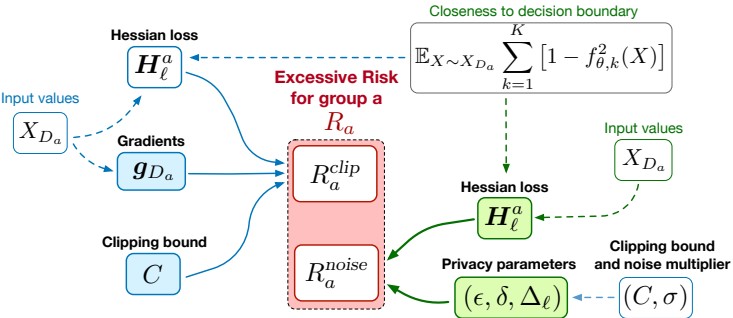

Figure 2: Diagram of the factors affecting the excessive risk $R_a$ for a group $a \in \mathcal{A}$ of individuals. Components affecting $R_a$ in output perturbation involve exclusively the green boxes while those affecting $R_a$ in DP-SGD involve both green and blue boxes. The main *direct* factors (e.g., those appearing in Eq. (4)) affecting the excessive risk clipping $R_a^{clip}$ and noise $R_a^{noise}$ components are highlighted within colored boxes. These direct factors are also regulated by *latent* factors, shown in white boxes, with dotted lines illustrating dependencies.

observation that the clipping value $C$ regulates the dominance of a factor over the other. Indeed, for sufficiently large (small) $C$ values $R_a^{noise}$ will dominate (be dominated by) $R_a^{clip}$.[3]

## 7  Why gradient clipping causes unfairness?

As highlighted above, there are three factors influencing the clipping effect to the excessive risk $R_a^{clip}$: the *Hessian loss*, the *gradient values*, and the *clipping bound*. This section illustrates their dependencies with the excessive risk, provides conditions to compare the disparate impacts between different groups, and shows the presence of an extra (latent) factor: the norm of the *input values* $X_{D_a}$, which plays a role to this disparate impacts by indirectly controlling the norms of gradient $g_{D_a}$ (see the diagram illustrated in Figure 2).

The next results assume that the empirical loss function $\mathcal{L}(\theta; D_a)$, associated with each group $a \in \mathcal{A}$, is convex and $\beta_a$-smooth. The analysis also consider learning rates $\eta \leq 1/\max_a \beta_a$ and gradients $g(B)$ and $\bar{g}(B)$ with small variances. Note that this is not restrictive as the variance decreases as a function of the batch size $B$. Finally, for notational convenience, and w.l.o.g., the result focus on the case in which $|\mathcal{A}| = 2$. As shown in the empirical assessment (see Appendix C), however, the conclusions carry on even in cases when the above assumptions may not hold.

**Theorem 3.** *Let $p_z = |D_z|/|D|$ be the fraction of training samples in group $z \in \mathcal{A}$. For groups $a, b \in \mathcal{A}$, $R_a^{clip} > R_b^{clip}$ whenever:*

$$\|g_{D_a}\| \frac{p_a^2}{2} \geq \frac{5}{2} C + \|g_{D_b}\| \left( 1 + p_b + \frac{p_b^2}{2} \right). \quad (5)$$

Theorem 3 provides a sufficient condition for which a group may have larger excessive risk than another solely based on the clipping term analysis. *It relates unfairness with the average (non-private) gradient norms of the groups $g_{D_a}$ and $g_{D_b}$ and the clipping value $C$.* As shown in the diagram of Figure 2, this result relates two main factors to the excessive risk due to clipping $R_a^{clip}$: **(1)** the *clipping bound $C$*, and **(2)** the (norm of the) *gradients* $\|g_{D_a}\|$. While the *relative dataset size* $p_a = |D_a|/|D|$ of each group also appears in Equation (5), our extensive experiments showed that this factor may not play a prime role in controlling the disparate impacts (see Appendix C).

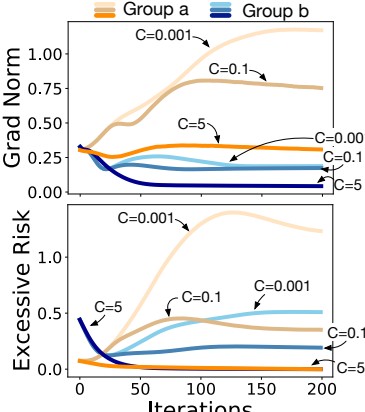

Figure 3: Impact of gradient clipping on gradient norms for different clipping bounds. Bank dataset.

The relation with these two factors is illustrated in Figure 3, which shows the impact of gradient clipping (for different $C$ values) to the gradient norms (top) and to the excessive risk $R_a$ (bottom). It

---

[3]This observation relates with the bias-variance trade-off typically observed in DP-SGD [25].

shows that the gradient norms reduce as $C$ increases and that the group with larger gradient norms have also larger excessive risk.

Finally, the diagram in Figure 2 also shows the presence of an additional factor affecting the gradient norms: *the input norms*, whose average is denoted $X_{D_a} = \mathbb{E}_{X \sim X_{D_a}} \|X\|$, in the figure. While this aspect is not directly evident in Theorem 3, the following examples highlight the positive correlation between input and gradients norms when considering a linear classifier and a feedforward neural network.

**Example 1.** *Consider the ERM problem* (L) *for a linear classifier* $f_{\boldsymbol{\theta}}(X) \stackrel{def}{=} \mathrm{softmax}(\boldsymbol{\theta}^T X)$ *and cross-entropy loss* $\ell(f_{\boldsymbol{\theta}}(X), Y) = -\sum_{i=1}^K Y_i \log f_{\boldsymbol{\theta}}^i(X)$ *where* $K$ *is the number of classes. The gradient of the loss function at a given data point* $(X, Y)$ *is:* $\boldsymbol{g}_X = \nabla\boldsymbol{\theta}\ell(f_{\boldsymbol{\theta}}(X), Y) = (\boldsymbol{Y} - \boldsymbol{f}) \otimes X$. *The result is by [7] and it suggests that the gradient norms are proportional to the input norms:* $\|\boldsymbol{g}_X\| \propto \|X\|$.

**Example 2.** *Next, consider a neural network with single hidden layer,* $f_{\boldsymbol{\theta}}(X) \stackrel{def}{=} \mathrm{softmax}\left(\boldsymbol{\theta}_1^T o(\boldsymbol{\theta}_2^T X)\right)$, *where* $o(\cdot)$ *is a proper activation function and* $\boldsymbol{\theta}_1, \boldsymbol{\theta}_2$ *are the model parameters. It can be seen that* $\|\boldsymbol{g}_X\| \propto \|\nabla_{\boldsymbol{\theta}_1}\ell(f_{\boldsymbol{\theta}}(X), Y)\| + \|\nabla_{\boldsymbol{\theta}_2}\ell(f_{\boldsymbol{\theta}}(X), Y)\|$, *where* $\|\nabla_{\boldsymbol{\theta}_2}\ell(f_{\boldsymbol{\theta}}(X), Y)\| \propto \|X\|$. *The full derivations are reported in Appendix D.*

Both examples illustrate a correlation between the gradients norms $\|\boldsymbol{g}_X\|$ and input norms $\|X\|$ for a given data sample $X$. This behavior is also illustrated in Figure 4, which highlights a positive correlation between the individual inputs and the gradients norms obtained while privately training a simple neural network (with one hidden layer) using DP-SGD on the Bank dataset. The experiment use $C = 0.1$ and $\sigma = 1$. The correlation decreases during training since the gradients norms reduce as training advances.

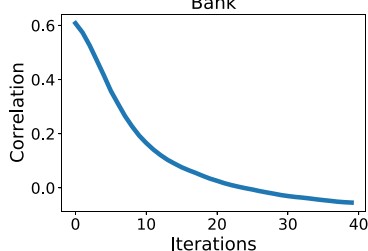

Figure 4: Correlation between inputs and gradients norms.

*These observations imply that group data with large input norms—typically defining the tail of data distribution—result in large gradient norms and, thus, as shown in Theorem 3, may have larger disproportionate impacts than groups with smaller input norms, under DP-SGD.* This analysis is in alignment with the empirical observation raised in [3], showing that samples at the tail of a distribution may experience larger accuracy losses, in private training, with respect to other samples.

While the above shows a dependency between gradients and clipping bound, as illustrated in the ($R_a^{clip}$) equation, the group excessive risk is also affected by the Hessian values. However, as shown in Appendix C, the Hessian factor is almost always dominated by the other factors examined in this section. This is due to the presence of the multiplier $\eta^2/2$ which attenuate the impact of the Hessian value to the excessive risk due to clipping in conjunction with the smoothness assumptions, which prevents the Hessian values to grow too large.

In summary, the main factors affecting $R_a^{clip}$ for a group $a \in \mathcal{A}$ are the norm of the group gradients $\boldsymbol{g}_{D_a}$, in turn controlled by the norm of the inputs $X_{D_a}$, and the clipping bound $C$.

## 8 Why noise addition causes unfairness?

Next, the paper analyzes the factors influencing the noise effect to the excessive risk $R_a^{noise}$, which, as highlighted in Theorem 2, for DP-SGD and Theorem 1 for output perturbation, are the *Hessian loss*, and the *privacy loss parameters* $(\epsilon, \delta, \Delta_\ell)$ (see also Figure 2). Noting that the privacy parameters have a multiplicative effect on the Hessian loss (see Equations ($R_a^{noise}$) and (3)), the following analysis, treats them as constants, and restricts focus on the effects of the Hessian trace to the disparate impacts.

The following result provide a condition to compare the disparate impacts between different groups,

**Theorem 4.** *For groups* $a, b \in \mathcal{A}$, $R_a^{noise} > R_b^{noise}$ *whenever*

$$\mathrm{Tr}(H_\ell^a) > \mathrm{Tr}(H_\ell^b).$$

Note the connection of the result above with Theorem 1. Additionally, as illustrated in the diagram of Figure 2 the Hessian trace for a group is controlled by two (latent) factors: **(1)** The average distance of the group data to the decision boundary, and **(2)** The values of the group input norms. While

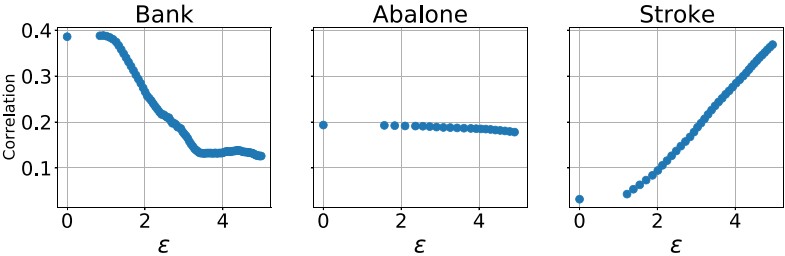

Figure 6: Correlation between input norms and excessive risk; DP-SGD with $C = 0.1$ and $\sigma = 1.0$.

these aspects are not directly evident in Theorem 4, the following highlights the positive correlation between these two factors and the Hessian Traces.

**Example 3.** *Consider the same setting of Example 1. The Hessian of the cross entropy loss of a sample $X \sim D$ is given by $H_\ell^X = [(diag(\boldsymbol{f}) - \boldsymbol{f}\boldsymbol{f}^T) \otimes XX^T]$, where $\otimes$ is the Kronecker product [7]. This result suggests that the trace of the Hessian for sample $X$ is proportional to its input norm: $\mathrm{Tr}(H_\ell^X) \propto \|X\|^2$. Additionally it also shows that: $\mathrm{Tr}(H_\ell^X) \propto (1 - \sum_{k=1}^K \boldsymbol{f}_{\boldsymbol{\theta},k}^2(X))$, where $K$ is the number of classes, whose term is connected to the distance to the decision boundary, as shown next.*

The following result highlights the connection between the term $(1 - \sum_{k=1}^K \boldsymbol{f}_{\boldsymbol{\theta},k}^2(X))$ and the distance of sample $X$ to the decision boundary.

**Theorem 5.** *Consider a $K$-class classifier $\boldsymbol{f}_{\boldsymbol{\theta},k}$ ($k \in [K]$). For a given sample $X \sim D$, the term $\left(1 - \sum_{k=1}^K \boldsymbol{f}_{\boldsymbol{\theta},k}^2(X)\right)$ is maximized when $\boldsymbol{f}_{\boldsymbol{\theta},k}(X) = 1/\kappa$ and minimized when $\exists k \in [K]$ s.t. $\boldsymbol{f}_{\boldsymbol{\theta},k}(X) = 1$ and $\boldsymbol{f}_{\boldsymbol{\theta},k'} = 0 \; \forall k' \in [K], k \neq k$.*

That is, the term $\left(1 - \sum_{k=1}^K \boldsymbol{f}_{\boldsymbol{\theta},k}^2(X)\right)$ is maximized (minimized) when the sample $X$ is close (far) to the decision boundary. Since, as shown in Example 3 this term can be proportional to the Hessian trace, then the aforementioned relation also indicates a connection between the Hessian trace value for a sample and its distance to the decision boundary: The closest (farther) is a sample $X$ to the decision boundary the larger (smaller) is the associated Hessian trace value $\mathrm{Tr}(\boldsymbol{H}_\ell^X)$.

This is intuitive as the model decision are less robust to the presence of noise in the model (e.g., as that introduced by a DP mechanism) for the samples which are close to the decision boundary w.r.t. those which are far from it.

An analogous behavior is also observed in Neural Networks and described in Appendix D due to space constraints. Figure 5 illustrates this behavior using the same setting adopted in Figure 4. It highlights the positive correlation between the input norm, the trace of Hessian, and the closeness to the decision boundary for a given sample $X$.

While the above discusses the relation between input norms and Hessian losses, Figure 6 illustrates this dependencies with the excessive risk, which is one of the main objective of the analysis, on three datasets. *Once again this observation recognizes the difference in input norms as a crucial proxy to unfairness: Groups with larger input norms will tend to have larger disproportionate impacts under private training than groups with smaller input norms.*

Figure 5: Correlation between trace of Hessian with closeness to boundary (dark color) and input norm (light color).

In summary, the main factor affecting $R_a^{noise}$ for a group $a \in \mathcal{A}$ is the Hessian loss $\boldsymbol{H}_\ell^a$, which, in turn, is controlled by the group's distance to the decision boundary and by their inputs norm.

## 9   Mitigation solution

The previous sections showed that, in DP-SGD, the excessive risk $R_a$ for a group $a \in \mathcal{A}$ could be decomposed into two factors $R_a^{clip}$, due to clipping, and $R_a^{noise}$, due to noise addition. In turn, it identified the gradients values $\boldsymbol{g}_{D_a}$ associated with the samples of group $a$ and the clipping bound $C$

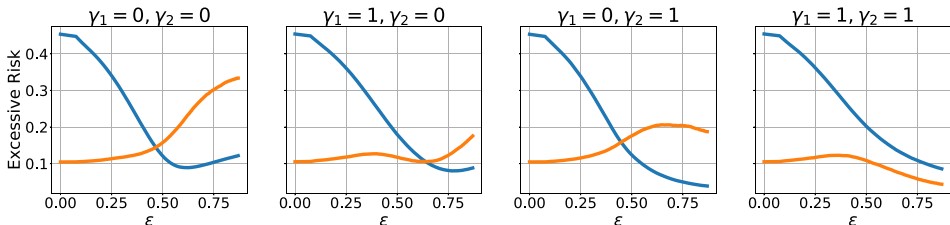

Figure 7: Mitigating solution: Excessive risk gap at varying of the privacy loss $\epsilon$ on the Bank dataset for different values of $\gamma_1$ and $\gamma_2$. Majority (minority) group is shown in dark (light) colors.

as the main sources of disparate impact in component $R_a^{clip}$, and the (trace of the) Hessian $\boldsymbol{H}_\ell^a$ of the group $a$ loss function as the main source of disparate impact in component $R_a^{noise}$.

A solution to mitigate the effects of these components to the excessive risk gap is to equalize the factors responsible for $R_a^{clip}$ and $R_a^{noise}$ among all group $a \in \mathcal{A}$ during private training. The resulting empirical risk loss becomes:

$$\min_{\boldsymbol{\theta}} \mathcal{L}(\boldsymbol{\theta}; D) + \sum_{a \in \mathcal{A}} \left( \gamma_1 \left| \langle \boldsymbol{g}_{D_a} - \boldsymbol{g}_D, \boldsymbol{g}_D - \bar{\boldsymbol{g}}_D \rangle \right| + \gamma_2 \left| \text{Tr}(\boldsymbol{H}_\ell^a) - \text{Tr}(\boldsymbol{H}_\ell) \right| \right), \tag{6}$$

where the component multiplied by $\gamma_1$ comes for simplifying the expression $|\langle \boldsymbol{g}_{D_a}, \boldsymbol{g}_D \rangle - \langle \boldsymbol{g}_{D_a}, \bar{\boldsymbol{g}}_D \rangle - \langle \boldsymbol{g}_D, \boldsymbol{g}_D \rangle - \langle \boldsymbol{g}_D, \bar{\boldsymbol{g}}_D \rangle|$ associated to the empirical risk gap $\xi_a$ of the main factor affecting $R_a^{clip}$, and component multiplied by $\gamma_2$ by the analogous expression for the main factor affecting $R_a^{noise}$. Note that this last component involves computing the Hessian matrices of the loss functions during each training step, which is a computationally expensive process. The previous section, however, showed a strong dependency between the trace of the Hessian losses and the distance to the decision boundary (Theorem 5). Thus, in place of Equation (6) the proposed mitigating solution solves:

$$\min_{\boldsymbol{\theta}} \mathcal{L}(\boldsymbol{\theta}; D) + \sum_{a \in \mathcal{A}} \left( \gamma_1 \left| \langle \boldsymbol{g}_{D_a} - \boldsymbol{g}_D, \boldsymbol{g}_D - \bar{\boldsymbol{g}}_D \rangle \right| + \gamma_2 \left| \mathbb{E}_{X \sim D_a} [1 - \sum_{k=1}^{K} f_{\theta,k}^2(X)] - \mathbb{E}_{X \sim D} [1 - \sum_{k=1}^{K} f_{\theta,k}^2(X)] \right| \right).$$

Figure 7 illustrates this approach at work, for various multipliers $\gamma_1$ and $\gamma_2$ on the Bank dataset with two protected group (blue = majority; orange = minority). Similar trends are shown for other datasets as well in Appendix C. The implementation uses a neural network with a single hidden layer and Suppose uses DP-SGD with $C = 0.1, \sigma = 5.0$. A clear trend arises: For appropriately selected values $\gamma_1$ and $\gamma_2$ the excessive risk gap between the majority and minority groups not only tends to be equalized, but it also decreases significantly for both groups. *These results imply that the proposed mitigating strategy may not only improve fairness but also the loss in utility of the private models.*

## 10  Limitations and conclusions

This work was motivated by the recent observations regarding the disparate impacts induced by DP in learning systems. The paper introduced a notion of fairness that relies on the concept of excessive risk, analyzed this fairness notion in output perturbation and DP-SGD for ERM problems, it isolated the relevant components related with noise addition and gradient clipping responsible for the disparate impacts, studied the main factors affecting these components, and introduced a mitigation solution.

This study recognizes the following limitations: Firstly, the analyses in Section 7 requires the ERM losses to be smooth and convex. While these are common assumptions adopted in the analysis of private ERM [29, 10], the generalization to the non-convex case is an interesting open question. The second limitation regards the selection of the multipliers $\gamma_1$ and $\gamma_2$ in Equation 6. While the paper does not investigate how to optimally selecting these values, the adoption of a Lagrangian Dual framework, as in [25], could a useful tool to the automatic selection of such parameters, for an extra privacy cost. Finally, the proposed mitigation solution negatively affects the training runtime and the design of more efficient solutions and implementations is an interesting challenge.

Despite these limitations, given the increasingly key role of differential privacy in machine learning, we believe that this work may represent an important and broadly useful step toward understanding the roots of the disparate impacts observed in differentially private learning systems.

## Acknowledgments and Disclosure of Funding

This research is partially supported by NSF grant 2133169. Its views and conclusions are those of the authors only.

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
