# Supplemental Material

## A   Missing Proofs

**Theorem 1.** *Let $\ell$ be a twice differentiable and convex loss function and consider the output perturbation mechanism described above. Then, the excessive risk gap for group $a \in \mathcal{A}$ is approximated by:*

$$\xi_a \approx \frac{1}{2}\Delta_\ell^2\sigma^2 \left| \mathrm{Tr}(\boldsymbol{H}_\ell^a) - \mathrm{Tr}(\boldsymbol{H}_\ell) \right|, \tag{3}$$

*where $\boldsymbol{H}_\ell^a = \nabla_{\boldsymbol{\theta}^*}^2 \sum_{(X,A,Y)\in D_a}\ell(f_{\boldsymbol{\theta}^*}(X), Y)$ is the Hessian matrix of the loss function at the optimal parameters vector $\boldsymbol{\theta}^*$, computed using the group data $D_a$, $\boldsymbol{H}^\ell$ is the analogous Hessian computed using the population data $D$, and $\mathrm{Tr}(\cdot)$ denotes the trace of a matrix.*

*Proof.* Recall that the output perturbation mechanism adds Gaussian noise directly to the non-private model parameters $\boldsymbol{\theta}^*$ to obtain the private parameters $\tilde{\boldsymbol{\theta}}$. Denote $\psi \sim \mathcal{N}(0, \boldsymbol{I}\Delta_\ell^2\sigma^2)$ the random noise vector with the same size as $\boldsymbol{\theta}^*$. Then $\tilde{\boldsymbol{\theta}} = \boldsymbol{\theta}^* + \psi$. Using a second order Taylor expansion around $\boldsymbol{\theta}^*$ the private risk function for group $a \in \mathcal{A}$ is approximated as follows:

$$\mathcal{L}(\tilde{\boldsymbol{\theta}}, D_a) = \mathcal{L}(\boldsymbol{\theta}^* + \psi, D_a) \approx \mathcal{L}(\boldsymbol{\theta}^*, D_a) + \psi^T\nabla_{\boldsymbol{\theta}^*}\mathcal{L}(\boldsymbol{\theta}^*, D_a) + \frac{1}{2}\psi^T\boldsymbol{H}_\ell^a\psi. \tag{7}$$

Taking the expectation with respect to $\psi$ on both sides of the above equation results in:

$$\mathbb{E}\left[\mathcal{L}(\tilde{\boldsymbol{\theta}}, D_a)\right] \approx \mathbb{E}\left[\mathcal{L}(\boldsymbol{\theta}^*, D_a)\right] + \mathbb{E}\left[\psi^T\nabla_{\boldsymbol{\theta}^*}\mathcal{L}(\boldsymbol{\theta}^*, D_a)\right] + \frac{1}{2}\mathbb{E}\left[\psi^T\boldsymbol{H}_\ell^a\psi\right] \tag{8a}$$

$$= \mathcal{L}(\boldsymbol{\theta}^*, D_a) + \frac{1}{2}\mathbb{E}\left[\psi^T\boldsymbol{H}_\ell^a\psi\right] \tag{8b}$$

$$= \mathcal{L}(\boldsymbol{\theta}^*, D_a) + \frac{1}{2}\sum_{i,j}\mathbb{E}\left[\psi_i(\boldsymbol{H}_\ell^a)_{ij}\psi_j\right] \tag{8c}$$

$$= \mathcal{L}(\boldsymbol{\theta}^*, D_a) + \frac{1}{2}\sum_{i}\mathbb{E}\left[\psi_i^2\right](\boldsymbol{H}_\ell^a)_{ii} \tag{8d}$$

$$= \mathcal{L}(\boldsymbol{\theta}^*, D_a) + \frac{1}{2}\Delta_\ell^2\sigma^2\,\mathrm{Tr}\left(\boldsymbol{H}_\ell^a\right), \tag{8e}$$

where equation (8b) follows from linearity of expectation, by observing that $\nabla_{\boldsymbol{\theta}^*}\mathcal{L}(\boldsymbol{\theta}^*, D_a)$ is a constant term, and that $\psi$ is a 0-mean noise variable, thus, $\mathbb{E}[\psi] = \boldsymbol{0}^T \times \nabla_{\boldsymbol{\theta}^*}\mathcal{L}(\boldsymbol{\theta}^*, D_a) = \boldsymbol{0}^T$. Equation (8c) follows by definition of Hessian matrix, where $(H_\ell^a)_{ij}$ denotes the entry with indices $i$ and $j$ of the matrix. Equation (8d) follows from that $\psi_i \perp \psi_j$, for all $i \neq j$, and Equation (8e) from that for a random variable $X$, $\mathbb{E}[X^2] = (\mathbb{E}[X])^2 + \mathrm{Var}[X]$, and $\mathrm{Var}[\psi_i] = \Delta_\ell^2\sigma^2\ \forall i$ and definition of Trace of a matrix.

Therefore, the group and population excessive risks are approximated as:

$$R_a(\boldsymbol{\theta}) = \mathbb{E}\left[\mathcal{L}(\tilde{\boldsymbol{\theta}}, D_a)\right] - \mathcal{L}(\boldsymbol{\theta}^*, D_a) \approx \frac{1}{2}\Delta_\ell^2\sigma^2\,\mathrm{Tr}\left(\boldsymbol{H}_\ell^a\right) \tag{9}$$

$$R(\boldsymbol{\theta}) = \mathbb{E}\left[\mathcal{L}(\tilde{\boldsymbol{\theta}}, D)\right] - \mathcal{L}(\boldsymbol{\theta}^*, D) \approx \frac{1}{2}\Delta_\ell^2\sigma^2\,\mathrm{Tr}\left(\boldsymbol{H}_\ell\right). \tag{10}$$

The claim follows by definition of excessive risk gap (Equation 2) subtracting Equation (9) from (10) in absolute values. $\square$

**Corollary 1.** *Consider the ERM problem for a linear model $f_{\boldsymbol{\theta}}(X) \overset{def}{=} \boldsymbol{\theta}^T X$, with $L_2$ loss function i.e., $\ell(f_{\boldsymbol{\theta}}(X), Y) = (f_{\boldsymbol{\theta}}(X) - Y)^2$. Then, output perturbation does not guarantee pure fairness.*

*Proof.* First, notice that for an $L_2$ loss function the trace of Hessian loss for a group $a \in \mathcal{A}$ is:

$$\mathrm{Tr}(\boldsymbol{H}_\ell^a) = \mathbb{E}_{x\sim D_a}\|X\|.$$

Therefore, from Theorem 1, the excessive risk gap $\xi_a$ for group $a$ is:

$$\xi_a \approx \frac{1}{2}\Delta_\ell^2\sigma^2 \left| \mathbb{E}_{x\sim D_a}\|X\| - \mathbb{E}_{x\sim D}\|X\| \right|. \tag{11}$$

Notice that $\xi_a$ is larger than zero only if the average input norm of group $a$ is different with that of the population one. Since this condition cannot be guaranteed in general, the output perturbation mechanism for a linear ERM model under the $L_2$ loss does not guarantee pure fairness. □

**Corollary 2.** *If for any two groups $a, b \in \mathcal{A}$ their average group norms $\mathbb{E}_{X_a\sim D_a}\|X_a\| = \mathbb{E}_{X_b\sim D_b}\|X_b\|$ have identical values, then output perturbation with $L_2$ loss function provides pure fairness.*

*Proof.* The above follows directly by observing that, when the average norms of any two groups have identical values, $\xi_a \approx 0$ for any group $a \in \mathcal{A}$ (see Equation (11)), and thus the average norm of each group also coincide with that of the population. □

The above indicates that as long as the average group norm is invariant across different groups, then output perturbation mechanism provides pure fairness.

**Theorem 2.** *Consider the ERM problem* (L) *with loss $\ell$ twice differentiable with respect to the model parameters. The expected loss $\mathbb{E}[\mathcal{L}(\theta_{t+1}; D_a)]$ of group $a\in\mathcal{A}$ at iteration $t+1$, is approximated as:*

$$\mathbb{E}\left[\mathcal{L}(\theta_{t+1}; D_a)\right] \approx \underbrace{\mathcal{L}(\theta_t; D_a) - \eta\langle g_{D_a}, g_D\rangle + \frac{\eta^2}{2}\mathbb{E}\left[g_B^T H_\ell^a g_B\right]}_{\textit{non-private term}} \tag{4}$$

$$+ \underbrace{\eta\left(\langle g_{D_a}, g_D\rangle - \langle g_{D_a}, \bar{g}_D\rangle\right) + \frac{\eta^2}{2}\left(\mathbb{E}\left[\bar{g}_B^T H_\ell^a \bar{g}_B\right] - \mathbb{E}\left[g_B^T H_\ell^a g_B\right]\right)}_{\textit{private term due to clipping}} \tag{$R_a^{clip}$}$$

$$+ \underbrace{\frac{\eta^2}{2}\text{Tr}(H_\ell^a)C^2\sigma^2}_{\textit{private term due to noise}} \tag{$R_a^{noise}$}$$

*where the expectation is taken over the randomness of the private noise and the mini-batch selection, and the terms $g_Z$ and $\bar{g}_Z$ denote, respectively, the average non-private and private gradients over subset $Z$ of $D$ at iteration $t$ (the iteration number is dropped for ease of notation).*

*Proof.* The proof of Theorem 2 relies on the following two second order Taylor approximations: **(1)** The first approximates the ERM loss at iteration $t + 1$ under non-private training, i.e., $\theta_{t+1} = \theta_t - \eta g_B$, where $B \subseteq D$ denotes the minibatch. **(2)** The second approximates expected ERM loss under private-training, i.e $\theta_{t+1} = \theta_t - \eta(\bar{g}_B + \psi)$ where $\psi \sim \mathcal{N}(0, IC^2\sigma^2)$. Finally, the result is obtained by taking the difference of these approximations under private and non-private training.

**1. Non-private term.** The non private term of Theorem 2 can be derived using second order Taylor approximation as follows:

$$\mathcal{L}(\theta_{t+1}, D_a) = \mathcal{L}(\theta_t - \eta g_B, D_a) \approx \mathcal{L}(\theta_t, D_a) - \eta\langle g_{D_a}, g_B\rangle + \frac{\eta^2}{2}g_B^T H_\ell^a g_B \tag{12}$$

Taking the expectation with respect to the randomness of the mini-batch $B$ selection on both sides of the above approximation, and noting that $\mathbb{E}[g_B] = g_D$ (as $B$ is selected randomly from dataset $D$), it follows:

$$\mathbb{E}[\mathcal{L}(\theta_{t+1}, D_a)] \approx \mathcal{L}(\theta_t, D_a) - \eta\mathbb{E}[\langle g_{D_a}, g_B\rangle] + \frac{\eta^2}{2}\mathbb{E}[g_B^T H_\ell^a g_B] \tag{13a}$$

$$= \mathcal{L}(\theta_t, D_a) - \eta\langle g_{D_a}, g_D\rangle + \frac{\eta^2}{2}\mathbb{E}[g_B^T H_\ell^a g_B]. \tag{13b}$$

**2. Private term (due to both clipping and noise).** Consider the private update in DP-SGD, i.e., $\theta_{t+1} = \theta_t - \eta(\bar{g}_B + \psi)$. Again, applying a second order Taylor approximation around $\theta_t$ allows us to estimate the expected private loss at iteration $t + 1$ as:

$$\mathcal{L}(\theta_{t+1}, D_a) = \mathcal{L}(\theta_t - \eta(\bar{g}_B + \psi), D_a)$$

$$\approx \mathcal{L}(\theta_t, D_a) - \eta \langle g_{D_a}, \bar{g}_B + \psi \rangle + \frac{\eta^2}{2}(\bar{g}_B + \psi)^T H_\ell^a (\bar{g}_B + \psi) \tag{14a}$$

$$= \mathcal{L}(\theta_t, D_a) - \eta \langle g_{D_a}, \bar{g}_B \rangle - \eta \langle g_{D_a}, \psi \rangle + \frac{\eta^2}{2} \bar{g}_B^T H_\ell^a \bar{g}_B \tag{14b}$$

$$+ \frac{\eta^2}{2} \left( \psi^T H_\ell^a \bar{g}_B + \bar{g}_B^T H_\ell^a \psi + \psi^T H_\ell^a \psi \right)$$

Taking the expectation with respect to the randomness of the mini-batch $B$ selection and with respect to the randomness of noise $\psi$ on both sides of the above equation gives:

$$\mathbb{E}[\mathcal{L}(\theta_{t+1}, D_a)] \approx \mathbb{E}\Big[\mathcal{L}(\theta_t, D_a) - \eta \langle g_{D_a}, \bar{g}_B \rangle - \eta \langle g_{D_a}, \psi \rangle + \frac{\eta^2}{2} \bar{g}_B^T H_\ell^a \bar{g}_B \tag{15a}$$

$$+ \frac{\eta^2}{2} \left( \psi^T H_\ell^a \bar{g}_B + \bar{g}_B^T H_\ell^a \psi + \psi^T H_\ell^a \psi \right) \Big]$$

$$= \mathcal{L}(\theta_t, D_a) - \eta \langle g_{D_a}, \bar{g}_B \rangle - \eta \langle g_{D_a}, \mathbb{E}[\psi] \rangle + \frac{\eta^2}{2} \mathbb{E}\left[ \bar{g}_B^T H_\ell^a \bar{g}_B \right] \tag{15b}$$

$$+ \frac{\eta^2}{2} \left( \mathbb{E}[\psi]^T H_\ell^a \bar{g}_B + \bar{g}_B^T H_\ell^a \mathbb{E}[\psi] + \mathbb{E}\left[ \psi^T H_\ell^a \psi \right] \right)$$

$$= \mathcal{L}(\theta_t, D_a) - \eta \langle g_{D_a}, \bar{g}_B \rangle + \frac{\eta^2}{2} \mathbb{E}\left[ \bar{g}_B^T H_\ell^a \bar{g}_B \right] + \frac{\eta^2}{2} \mathbb{E}\left[ \psi^T H_\ell^a \psi \right] \tag{15c}$$

$$= \mathcal{L}(\theta_t, D_a) - \eta \langle g_{D_a}, \bar{g}_B \rangle + \frac{\eta^2}{2} \mathbb{E}\left[ \bar{g}_B^T H_\ell^a \bar{g}_B \right] + \frac{\eta^2}{2} \operatorname{Tr}\left( H_\ell^a \right) C^2 \sigma^2, \tag{15d}$$

where (15b), and (15c) follow from linearity of expectation and from that $\mathbb{E}[\psi] = 0$, since $\psi$ is a 0-mean noise variable. Equation (15d) follows from that,

$$\mathbb{E}\left[ \psi^T H_\ell^a \psi \right] = \mathbb{E}\left[ \sum_{i,j} \psi_i (H_\ell^a)_{i,j} \psi_j \right] = \sum_i \mathbb{E}\left[ \psi_i^2 (H_\ell^a)_{i,i} \right] = \operatorname{Tr}\left( H_\ell^a \right) C^2 \sigma^2,$$

since $\mathbb{E}[\psi^2] = \mathbb{E}[\psi]^2 + \operatorname{Var}[\psi]$ and $\mathbb{E}[\psi] = 0$ while $\operatorname{Var}[\psi] = C^2 \sigma^2$.

Note that in the above approximation (Equation (15)), the component

$$\mathcal{L}(\theta_t, D_a) - \eta \langle g_{D_a}, \bar{g}_B \rangle + \frac{\eta^2}{2} \mathbb{E}\left[ \bar{g}_B^T H_\ell^a \bar{g}_B \right] \tag{16}$$

is associated to the SGD update step in which gradients have been clipped to the clipping bound value $C$, i.e. $\theta_{t+1} = \theta_t - \eta(\bar{g}_B)$.

Next, the component

$$\frac{\eta^2}{2} \operatorname{Tr}\left( H_\ell^a \right) C^2 \sigma^2 \tag{17}$$

is associated to the SGD update step in which the noise $\psi$ is added to the gradients.

If we take the difference between the approximation associated with the non-private loss term, obtained in Equation 13b, with that associated with the private loss term, obtained in Equation 15d, we can derive the effect of a single step of (private) DP-SGD compared to its non-private counterpart:

$$\mathbb{E}[\mathcal{L}(\theta_{t+1}; D_a)] \approx \mathcal{L}(\theta_t; D_a) - \eta \langle g_{D_a}, g_D \rangle + \frac{\eta^2}{2} \mathbb{E}\left[ g_B^T H_\ell^a g_B \right] \tag{18a}$$

$$+ \eta (\langle g_{D_a}, g_D \rangle - \langle g_{D_a}, \bar{g}_D \rangle) + \frac{\eta^2}{2} \left( \mathbb{E}\left[ \bar{g}_B^T H_\ell^a \bar{g}_B \right] - \mathbb{E}\left[ g_B^T H_\ell^a g_B \right] \right) \tag{18b}$$

$$+ \frac{\eta^2}{2} \operatorname{Tr}\left( H_\ell^a \right) C^2 \sigma^2. \tag{18c}$$

In the above,

- The components in Equation (18a) are associated with the loss under non-private training (see again Equation 13b);
- The components in Equation (18b) is associated with for excessive risk due to gradient clipping;
- Finally, the components in Equation (18c) is associated with the excessive risk due to noise addition.

□

Next, the paper proves Theorem 3. This result is based on the following assumptions.

**Assumption 1.** *[Convexity and Smoothness assumption] For a group $a \in \mathcal{A}$, its empirical loss function $\mathcal{L}(\theta, D_a)$ is convex and $\beta_a$-smooth.*

**Assumption 2.** *Let $B \subseteq D$ be a subset of the dataset $D$, and consider a constant $\varepsilon \geq 0$. Then, the variance associated with the gradient norms of a random mini-batch B, $\sigma_B^2 = \mathrm{Var}\left[\|g_B\|\right] \leq \varepsilon$ as well as that associated with its clipped counterpart, $\bar{\sigma}_B^2 = \mathrm{Var}\left[\|\bar{g}_B\|\right] \leq \varepsilon$.*

The assumption above can be satisfied when the mini-batch size is large enough. For example, the variance is 0 when $|B| = |D|$.

**Assumption 3.** *The learning rate used in DP-SGD $\eta$ is upper bounded by quantity $1/\max_{z \in \mathcal{A}} \beta_z$.*

**Theorem 3.** *Let $p_z = |D_z|/|D|$ be the fraction of training samples in group $z \in \mathcal{A}$. For groups $a, b \in \mathcal{A}$, $R_a^{clip} > R_b^{clip}$ whenever:*

$$\|g_{D_a}\| \frac{p_a^2}{2} \geq \frac{5}{2}C + \|g_{D_b}\| \left(1 + p_b + \frac{p_b^2}{2}\right). \tag{5}$$

To ease notation, the statement of the theorem above uses $\varepsilon = 0$ (See Assumption 2) but the theorem can be generalized to any $\varepsilon \geq 0$.

The following Lemmas are introduced to aid the proof of Theorem 3.

**Lemma 1.** *Consider the ERM problem* (L) *solved with DP-SGD with clipping value C. The following average clipped per-sample gradients $\bar{g}_Z$, where $Z \subseteq D$, has norm at most C.*

*Proof.* The result follows by triangle inequality:

$$\begin{aligned}
\|\bar{g}_{D_Z}\| &= \left\|\frac{1}{|D_Z|} \sum_{i \in D_Z} \bar{g}_i\right\| \\
&\leq \frac{1}{|D_Z|} \sum_{i \in D_Z} \|\bar{g}_i\| \\
&= \frac{1}{|D_Z|} \sum_{i \in D_Z} \left\|g_i \min\left(1, \frac{C}{\|g_i\|}\right)\right\| \\
&\leq \frac{1}{|D_Z|} \sum_{i \in D_Z} C = C.
\end{aligned}$$

□

The next Lemma derives a lower and an upper bound for the component $\mathbb{E}[\bar{g}_B^T H_\ell^a \bar{g}_B] - \mathbb{E}[g_B^T H_\ell^a g_B]$, which appears in the excessive risk term due to clipping $R_a^{clip}$ for some group $a \in \mathcal{A}$.

**Lemma 2.** *Consider the ERM problem* (L) *with loss $\ell$, solved with DP-SGD with clipping value C. Further, let $\varepsilon = 0$ (see Assumption 2). For any group $a \in \mathcal{A}$, the following inequality holds:*

$$-\beta_a \|g_D\|^2 \leq \mathbb{E}[\bar{g}_B^T H_\ell^a \bar{g}_B] - \mathbb{E}[g_B^T H_\ell^a g_B] \leq \beta_a C^2 \tag{19}$$

*Proof.* Consider a group $a \in \mathcal{A}$. By the convexity assumption of the loss function, the Hessian $H_\ell^a$ is a positive semi-definite matrix, i.e., for all real vectors of appropriate dimensions $v$, it follows that $v^T H_\ell^a v \geq 0$.

Therefore, for a subset $B \subseteq D$ the following inequalities hold:

- $\bar{g}_B H_\ell^a \bar{g}_B \geq 0$,
- $g_B^T H_\ell^a g_B \geq 0$ .

Additionally their expectations $\mathbb{E}[\bar{g}_B H_\ell^a \bar{g}_B]$ and $\mathbb{E}[g_B^T H_\ell^a g_B]$ are non-negative.

By the smoothness property of the loss function, $\bar{g}_B^T H_\ell^a \bar{g}_B \leq \beta_a \|\bar{g}_B\|^2$, thus:

$$\mathbb{E}[\bar{g}_B^T H_\ell^a \bar{g}_B] \leq \beta_a \mathbb{E}\left[\|\bar{g}_B\|^2\right] \tag{20a}$$

$$= \beta_a (\mathbb{E}[\|\bar{g}_B\|]^2 + \mathrm{Var}\left[\|\bar{g}_B\|\right]) \tag{20b}$$

$$\leq \beta_a (C^2 + \bar{\sigma}_B^2) \tag{20c}$$

$$\leq \beta_a (C^2 + \varepsilon), \tag{20d}$$

where Equation (20b) follows from that $\mathbb{E}[X^2] = (\mathbb{E}[X])^2 + \mathrm{Var}[X]$, Equation (20c) is due to Lemma 1, and finally, the last inequality is due to Assumption 2.

Therefore, since $\varepsilon = 0$ by assumption of the Lemma, the following upper bound holds:

$$\mathbb{E}[\bar{g}_B^T H_\ell^a \bar{g}_B] - \mathbb{E}[g_B^T H_\ell^a g_B] \leq \beta_a C^2. \tag{21}$$

Next, notice that

$$\mathbb{E}[\bar{g}_B^T H_\ell^a \bar{g}_B] - \mathbb{E}[g_B^T H_\ell^a g_B] \geq -\mathbb{E}[g_B^T H_\ell^a g_B] \tag{22a}$$

$$\geq -\mathbb{E}[\beta_a \|g_B\|^2] \tag{22b}$$

$$= -\beta_a \left(\mathbb{E}[\|g_B\|]^2 + \mathrm{Var}\left[\|g_B\|\right]\right) \tag{22c}$$

$$= -\beta_a \|g_D\|^2, \tag{22d}$$

where the inequality in Equation (22a) follows since both terms on the left hand side of the Equation are non negative. Equation (22b) follows by smoothness assumption of the loss function. Equation (22c) follows by definition of expectation of a random variable, since $\mathbb{E}[X]^2 = \mathbb{E}[X^2] + \mathrm{Var}[X]$. Finally, Equation (22d) follows from that $\mathrm{Var}[g_B] \leq \varepsilon = 0$ by Assumption 2, and that $\varepsilon = 0$ by assumption of the Lemma, and thus the norms $\|g_B\| = \|g_D\|$ and, thus, $\mathbb{E}[g_B] = g_D$. Therefore if follows:

$$-\beta_a \|g_D\|^2 \leq \mathbb{E}[\bar{g}_B^T H_\ell^a \bar{g}_B] - \mathbb{E}[g_B^T H_\ell^a g_B]. \tag{23}$$

which concludes the proof. □

Again, the above uses $\varepsilon = 0$ to simplify notation, but the results generalize to the case when $\varepsilon > 0$. In such a case, the bounds require slight modifications to involve the term $\varepsilon$.

**Lemma 3.** *Let $a, b \in \mathcal{A}$ be two groups. Consider the ERM problem* (L) *solved with DP-SGD with clipping value C and learning rate $\eta \leq 1/\max_{a \in \mathcal{A}} \beta_a$. Then, the difference on the excessive risk due to clipping $R_{clip}^a - R_{clip}^b$ is lower bounded as:*

$$R_{clip}^a - R_{clip}^b \geq \eta \left( \langle g_{D_a} - g_{D_b}, g_D - \bar{g}_D \rangle - \frac{1}{2} (\|g_D\|^2 + C^2) \right). \tag{24}$$

*Proof.* Recall that $B \subseteq D$ is the mini-batch during the resolution of DP-SGD. Using the lower and upper bounds obtained from Lemma 2, it follows:

$$R^a_{clip} - R^b_{clip} = \eta \left( \langle g_{D_a}, g_D \rangle - \langle g_{D_a}, \bar{g}_D \rangle \right) + \frac{\eta^2}{2} \left( \mathbb{E} \left[ \bar{g}_B^T H_\ell^a \bar{g}_B \right] - \mathbb{E} \left[ g_B^T H_\ell^a g_B \right] \right) \tag{25a}$$

$$- \eta \left( \langle g_{D_b}, g_D \rangle - \langle g_{D_b}, \bar{g}_D \rangle \right) - \frac{\eta^2}{2} \left( \mathbb{E} \left[ \bar{g}_B^T H_\ell^b \bar{g}_B \right] - \mathbb{E} \left[ g_B^T H_\ell^b g_B \right] \right)$$

$$= \eta \langle g_{D_a} - g_{D_b}, g_D - \bar{g}_D \rangle + \frac{\eta^2}{2} \left( \mathbb{E} \left[ \bar{g}_B^T H_\ell^a \bar{g}_B \right] - \mathbb{E} \left[ g_B^T H_\ell^a g_B \right] \right) \tag{25b}$$

$$- \frac{\eta^2}{2} \left( \mathbb{E} \left[ \bar{g}_B^T H_\ell^b \bar{g}_B \right] - \mathbb{E} \left[ g_B^T H_\ell^b g_B \right] \right)$$

$$\geq \eta \langle g_{D_a} - g_{D_b}, g_D - \bar{g}_D \rangle - \frac{\eta^2}{2} \beta_a \|g_D\|^2 - \frac{\eta^2}{2} \beta_b C^2 \tag{25c}$$

$$\geq \eta \langle g_{D_a} - g_{D_b}, g_D - \bar{g}_D \rangle - \frac{\eta^2}{2} \max_{z \in \mathcal{A}} \beta_z (\|g_D\|^2 + C^2) \tag{25d}$$

$$\geq \eta \left( \langle g_{D_a} - g_{D_b}, g_D - \bar{g}_D \rangle - \frac{1}{2} (\|g_D\|^2 + C^2) \right), \tag{25e}$$

where the inequality (25c) follows as a consequence of Lemma 2, and the inequality (25e) since $\eta \leq \frac{1}{\max_{a \in \mathcal{A}} \beta_a}$. $\square$

*Proof of Theorem 3.* We want to show that $R^a_{clip} > R^b_{clip}$ given Equation (5). Since, by Lemma 3 the difference $R^a_{clip} - R^b_{clip}$ is lower bounded – see Equation (24), the following shows that the right hand side of Equation (24) is positive, that is:

$$\langle g_{D_a} - g_{D_b}, g_D - \bar{g}_D \rangle - \frac{1}{2} \left( \|g_D\|^2 + C^2 \right) > 0. \tag{26}$$

First, observe that the gradients at the population level can be expressed as a combination of the gradients of the two groups $a$ and $b$ in the dataset: $g_D = p_a g_{D_a} + p_b g_{D_b}$ and $\bar{g} = p_a \bar{g}_{D_a} + p_b \bar{g}_{D_b}$.

By algebraic manipulation, and the above, Equation (26) can thus be expressed as:

$$(26) = \langle g_{D_a} - g_{D_b}, p_a g_{D_a} + p_b g_{D_b} - p_a \bar{g}_{D_a} - p_b \bar{g}_{D_b} \rangle - \frac{1}{2} (\|g_{D_a} p_a + g_{D_b} p_b\|^2 + C^2) \tag{27a}$$

$$= (p_a \|g_{D_a}\|^2 + p_b g_{D_a}^T g_{D_b} - p_a g_{D_a}^T \bar{g}_{D_a} - p_b g_{D_a}^T \bar{g}_{D_b} - p_a g_{D_b}^T g_{D_a} - p_b \|g_{D_b}\|^2 \tag{27b}$$

$$+ p_a g_{D_b}^T \bar{g}_{D_a} + p_b g_{D_b}^T \bar{g}_{D_b} - \frac{1}{2} (p_a^2 \|g_{D_a}\|^2 + 2 p_a p_b g_{D_a} g_{D_b} + p_b^2 \|g_{D_b}\|^2 + C^2).$$

Noting that for any vector $x, y$ the following inequality hold: $x^T y \geq -\|x\|\|y\|$, all the inner products in the above expression can be replaced by their lower bounds:

$$(26) \geq \|g_{D_a}\| \left( \|g_{D_a}\| p_a (1 - \frac{p_a}{2}) - p_b \|g_{D_b}\| - p_a C - p_b C - p_a \|g_{D_b}\| - p_a p_b \|g_{D_b}\| \right) \tag{28a}$$

$$- \|g_{D_b}\| - p_a p_b \|g_{D_b}\| \left( \|g_{D_b}\| p_b (1 + \frac{p_b}{2}) + p_a C + p_b C \right) - \frac{1}{2} C^2$$

$$= \|g_{D_a}\| \left( \|g_{D_a}\| p_a (1 - \frac{p_a}{2}) - p_a p_b \|g_{D_b}\| (p_b + p_a)(\|g_{D_b}\| + C) \right) \tag{28b}$$

$$- \|g_{D_b}\| \left( \|g_{D_b}\| p_b (1 + \frac{p_b}{2}) + (p_a + p_b) C \right) - \frac{1}{2} C^2$$

$$= \|g_{D_a}\| \left( \|g_{D_a}\| p_a (1 - \frac{p_a}{2}) - p_a p_b \|g_{D_b}\| - \|g_{D_b}\| - C \right) - \|g_{D_b}\| \left( \|g_{D_b}\| p_b (1 + \frac{p_b}{2}) + C \right) - \frac{1}{2} C^2 \tag{28c}$$

where the last equality is because $p_a + p_b = 1$, by assumption of the dataset having exactly two groups.

By theorem assumption, $\|\boldsymbol{g}_{D_a}\|\frac{p_a^2}{2} \geq \frac{5}{2}C + \|\boldsymbol{g}_{D_b}\|(1 + p_b + \frac{p_b^2}{2})$. It follows that $\|\boldsymbol{g}_{D_a}\| > \|\boldsymbol{g}_{D_b}\|$ and $\|\boldsymbol{g}_{D_a}\| > C$. Combined with Equation (28c) it follows that:

$$(28c) = \|\boldsymbol{g}_{D_a}\|\left(\|\boldsymbol{g}_{D_a}\|p_a(1 - \frac{p_a}{2}) - p_a p_b\|\boldsymbol{g}_{D_b}\| - \|\boldsymbol{g}_{D_b}\| - C - \|\boldsymbol{g}_{D_b}\|p_b(1 + \frac{p_b}{2}) - C\right) - \frac{1}{2}C^2 \quad (29a)$$

$$\geq \|\boldsymbol{g}_{D_a}\|\left(\|\boldsymbol{g}_{D_a}\|p_a(1 - \frac{p_a}{2}) - p_a p_b\|\boldsymbol{g}_{D_a}\| - 2C - \|\boldsymbol{g}_{D_b}\|(1 + p_b + \frac{p_b^2}{2})\right) - \frac{1}{2}C^2 \quad (29b)$$

$$\geq \|\boldsymbol{g}_{D_a}\|\left(\|\boldsymbol{g}_{D_a}\|p_a(1 - \frac{p_a}{2} - p_b) - 2C - \|\boldsymbol{g}_{D_b}\|(1 + p_b + \frac{p_b^2}{2})\right) - \frac{1}{2}C^2 \quad (29c)$$

$$= \|\boldsymbol{g}_{D_a}\|\left(\|\boldsymbol{g}_{D_a}\|\frac{p_a^2}{2} - 2C - \|\boldsymbol{g}_{D_b}\|(1 + p_b + \frac{p_b^2}{2})\right) - \frac{1}{2}C^2 \quad (29d)$$

$$\geq \|\boldsymbol{g}_{D_a}\|\frac{C}{2} - \frac{1}{2}C^2 \quad (29e)$$

$$> 0, \quad (29f)$$

where the last equality is because $\|\boldsymbol{g}_{D_a}\| > C$. □

**Theorem 4.** *For groups $a, b \in \mathcal{A}$, $R_a^{noise} > R_b^{noise}$ whenever*

$$\mathrm{Tr}(\boldsymbol{H}_\ell^a) > \mathrm{Tr}(\boldsymbol{H}_\ell^b).$$

*Proof.* Suppose $\mathrm{Tr}(\boldsymbol{H}_\ell^a) > \mathrm{Tr}(\boldsymbol{H}_\ell^b)$. By definition of $R_a^{noise}$ and $R_b^{noise}$ from Theorem 2 it follows that:

$$R_a^{noise} = \frac{\eta^2}{2}\,\mathrm{Tr}(\boldsymbol{H}_\ell^a)C^2\sigma^2 > \frac{\eta^2}{2}\,\mathrm{Tr}(\boldsymbol{H}_\ell^b)C^2\sigma^2 = R_b^{noise},$$

which concludes the proof.

□

**Theorem 5.** *Consider a K-class classifier $\boldsymbol{f}_{\theta,k}$ ($k \in [K]$). For a given sample $X \sim D$, the term $\left(1 - \sum_{k=1}^K \boldsymbol{f}_{\theta,k}^2(X)\right)$ is maximized when $\boldsymbol{f}_{\theta,k}(X) = 1/K$ and minimized when $\exists k \in [K]$ s.t. $\boldsymbol{f}_{\theta,k}(X) = 1$ and $\boldsymbol{f}_{\theta,k'} = 0 \; \forall k' \in [K], k' \neq k$.*

*Proof.* Fix an input $X$ of $D$ and denote $y_k = \boldsymbol{f}_{\theta,k}(X) \in [0, 1]$. Recall that $y_k$ represents the likelihood of the prediction of input $X$ to be associated with label $k$.

Note that, by Cauchy–Schwarz inequality

$$1 - \sum_{k=1}^K y_k^2 \leq 1 - K\left(\frac{\sum_i^K y_k}{K}\right)^2 \quad (30a)$$

$$= 1 - \frac{1}{K}, \quad (30b)$$

where Equation (30b) follows since $\sum_i^K y_k(X) = 1$. The above expression is maximized when

$$y_k = \boldsymbol{f}_{\theta,k}(X) = \frac{1}{K}.$$

Additionally, since $y_k \in [0, 1]$ it follows that $y_k^2 \leq y_k$. Hence,

$$1 - \sum_{k=1}^K y_k^2 \geq 1 - \sum_{i=1}^K y_k = 0. \quad (31)$$

To hold, the equality above, it must exists $k \in [K]$ such that $y_k = \boldsymbol{f}_{\theta,k}(X) = 1$ and for any other $k' \in [K]$ with $k' \neq k$, $y_{k'} = \boldsymbol{f}_{\theta,k'} = 0$. □

Given the connection of the term $1 - \sum_{k=1}^K (1 - f_{\theta,k}^2(X))$ and the associated (trace of the) Hessian loss $\boldsymbol{H}_f$, the result above suggests that the trace of the Hessian is minimized (maximized) when the classifier is very confident (uncertain) about the prediction of $X \sim D$, i.e., when $X$ is far (close) to the decision boundary.

## B   Experimental settings

**Datasets**   The paper uses the following UCI datasets to support its claims:

1. **Adult** (Income) dataset, where the task is to predict if an individual has low or high income, and the group labels are defined by race: *White* vs *Non-White* [6].

2. **Bank** dataset, where the task is to predict if a user subscribes a term deposit or not and the group labels are defined by age: *people whose age is less than 60 years old vs the rest* [21].

3. **Wine** dataset, where the task is to predict if a given wine is of good quality, and the group labels are defined by wine color: *red vs white* [6].

4. **Abalone** dataset, where the task is to predict if a given abalone ring exceeds the median value, and the group labels are defined by gender: *female vs male* [6].

5. **Parkinsons** dataset, where the task is to predict if a patient has total UPDRS score that exceeds the median value, and the group labels are defined by gender: *female vs male* [20].

6. **Churn** dataset, where the task is to predict if a customer churned or not. The group labels are defined by on gender: *female vs male* [12].

7. **Credit Card** dataset, where the task is to predict if a customer defaults a loan or not. The group labels are defined by gender: *female vs male* [8].

8. **Stroke** dataset, where the task is to predict if a patient have had a stroke based on their physical conditions. The group labels are defined by gender: *female vs male* [1].

All datasets were processed by standardization so each feature has zero mean and unit variance.

**Settings**   For output perturbation, the paper uses a Logistic regression model to obtain the optimal model parameters (we set the regularization parameter $\lambda = 1$) and add Gaussian noise to achieve privacy. The standard deviation of the noise required to the mechanism is determined following Balle and Wang [4].

For DP-SGD, the paper uses a neural network with single hidden layer with *tanh* activation function for the different datasets. The batch size $|B|$ is fixed to 32 and the learning rate $\eta = 1e - 4$. Unless specified we set the clipping bound $C = 0.1$ and noise multiplier $\sigma = 5.0$. The experiments consider 100 runs of DP-SGD with different random seeds for each configuration. We employ the Tensorflow Privacy toolbox to compute the privacy loss $\epsilon$ spent during training.

**Computing infrastructure**   All experiments were performed on a cluster equipped with Intel(R) Xeon(R) Platinum 8260 CPU @ 2.40GHz and 8GB of RAM.

**Software and libraries**   All models and experiments were written in Python 3.7 and in Pytorch 1.5.0.

**Code**   The code used for this submission is attached as supplemental material. All implementation of the experiments and proposed mitigation solution will be released upon publication.

## C   Additional experiments

### C.1   More on "Warm up: output perturbation"

**Correlation between Hessian trace and excessive risk**   The following provides additional empirical support for the claims of the main paper: *Groups with larger Hessian trace tend to have larger excessive risks* in this subsection.

The experiments in this sub-section use output perturbation. Figure 8 reports the excessive risk and Hessian traces for the two groups defined in the datasets (as described in Section B. The figure clearly illustrates that the groups with larger Hessian traces have larger excessive risk (i.e., experienced more unfairness) under private output perturbation when compared with the groups with smaller Hessian traces. These empirical findings are again a strong support for the claims of Theorem 1.

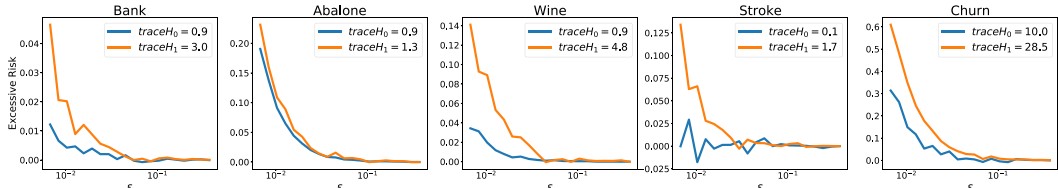

Figure 8: Correlation between excessive risk gap and Hessian Traces at varying of the privacy loss $\epsilon$.

**Impact of data normalization by group**  The next results provide evidence to support the following claim raised in Section 5: *Given the impact of gradient norms to unfairness, normalizing data independently for each group can help improve fairness*. Figure 9 shows the evolution of the excessive risk $R_a$ and $R_b$ for the dataset groups during training. The top plots present the results with standard data normalization (e.g., each sample data is normalized independently from its group membership) while the bottom plots show the counterpart results for models trained when the data was normalized within the group datasets $D_a$ and $D_b$. Note that the normalization adopted ensures that the data is 0-mean and of unit variance in each group dataset, which is a required condition to achieve the desired property.

The results clearly show that this strategy can not only reduce unfairness, but also the excessive risk gaps.

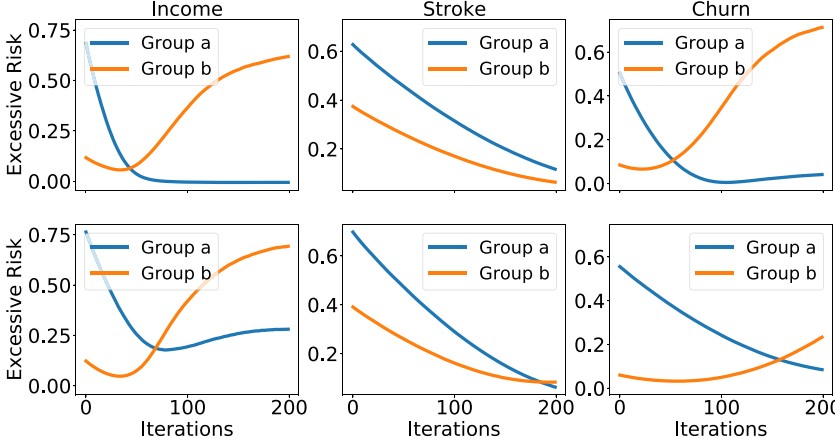

Figure 9: Excessive risk for each group without group normalization (top) and with group normalization (bottom).

## C.2  More on "Why gradient clipping causes unfairness?"

This section provides additional empirical evidence to support the claim made in Section 7 specifying the three direct factors influencing the clipping effect to the excessive risk: **(1)** the Hessian loss, **(2)** the gradient values, and **(3)** the clipping bound. Among these three factors, the gradient values and clipping bound are the dominant ones.

**Impact of gradient values and clipping bound** $C$  Figure **??** provides the relation between the gradient norm and the different choices of clipping bounds to the excessive risks. The results are shown for the Abalone, Churn and Credit Card datasets. The experiments show that gradient norms reduce as $C$ increases and that the group with larger gradient norms have also larger excessive risk. Similar results were achieved for other datasets as well (not reported to avoid redundancy).

**The Hessian loss is a minor impact factor to the excessive risk.**  As showed in the main text, the excessive risk associated to the gradient clipping for a particular group $a \in \mathcal{A}$ can be decomposed as:

$$R_a^{clip} = \eta \left( \langle g_{D_a}, g_D \rangle - \langle g_{D_a}, \bar{g}_D \rangle \right) + \frac{\eta^2}{2} \left( \mathbb{E}\left[ \bar{g}_B^T H_\ell^a \bar{g}_B \right] - \mathbb{E}\left[ g_B^T H_\ell^a g_B \right] \right) \tag{32}$$

Denote $\psi_a = \left( \mathbb{E}\left[ \bar{g}_B^T H_\ell^a \bar{g}_B \right] - \mathbb{E}\left[ g_B^T H_\ell^a g_B \right] \right)$. This quantity clearly depends on the Hessian loss $H_\ell^a$. However, under the assumptions in Theorem 3: convexity and smoothness of the loss function and the magnitude of the learning rate (i.e., that is small enough), the term $\psi_a$ will be a negligible component in $R_a^{clip}$.

While this is evident under those assumption, our empirical analysis has reported a similar behavior for loss function for which those conditions do not generally apply. In the following experiment we run DP-SGD on a neural network with single hidden layer and tracked the values of $R_a^{clip}$ and $\psi_a$ for each group $a \in \mathcal{A}$ during private training. These values are reported in Figure 10 for different datasets. It can be seen that the components $\psi_a$ (dotted lines) constitute a negligible amount to the excessive risk under gradient clipping $R_a^{clip}$.

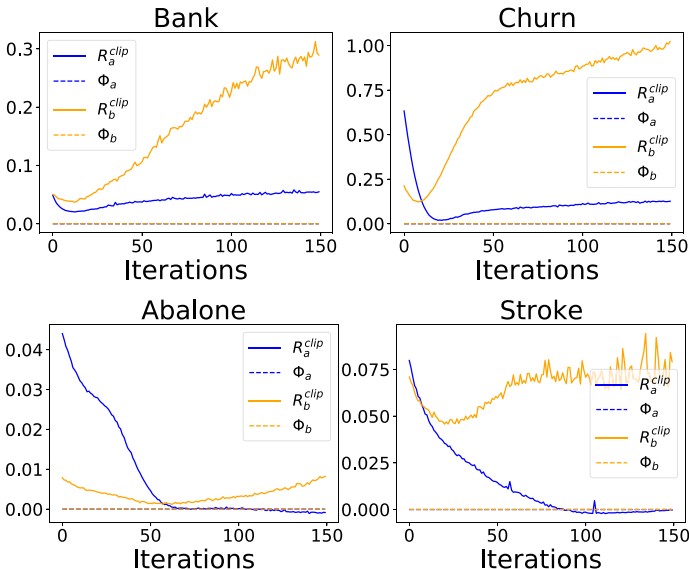

Figure 10: Values of $R_a^{clip}$ and $\psi_a$ during private training for a neural network classifier.

**Relative group data size is a minor impact factor to the excessive risk.**  Section 7 also observed that the relative group data size, $p_b/p_a$ for two groups $a, b \in \mathcal{A}$ had a minor impact on unfairness. Figure 11 provides empirical evidence to support this observation. It shows the effects of varying the relative group data $p_b/p_a$ to the gradient norms (top rows) and excessive risk (bottom rows) in three datasets: Abalone, Bank, and Income. The different relative group data ratios were obtained through subsampling. Notice that changing the relative group sizes does not result in a noticeable effect in the group gradient norms and excessive risk. These experiments demonstrate that the relative group data size might play a minor role in affecting unfairness.

These observation are also in alignment with the those raised by Farrand et al. [17], who showed that the disparate impact of DP on model accuracy is not limited to highly imbalanced data and can occur in situations where the groups are slightly imbalanced.

## C.3  More on "Why noise addition causes unfairness?"

Figure **??** illustrates the connection between the trace of the Hessian of the loss function at some sample $X \in D$ and its distance to the decision boundary. The figure clearly show that the closest (father) is a sample $X$ to the decision boundary, the larger (smaller) is the associated Hessian trace value $\mathrm{Tr}(H_\ell^X)$. The experiments are reported for datasets Parkinson, Stroke, Wine, and Churn, but once again they extend to other datasets as well.

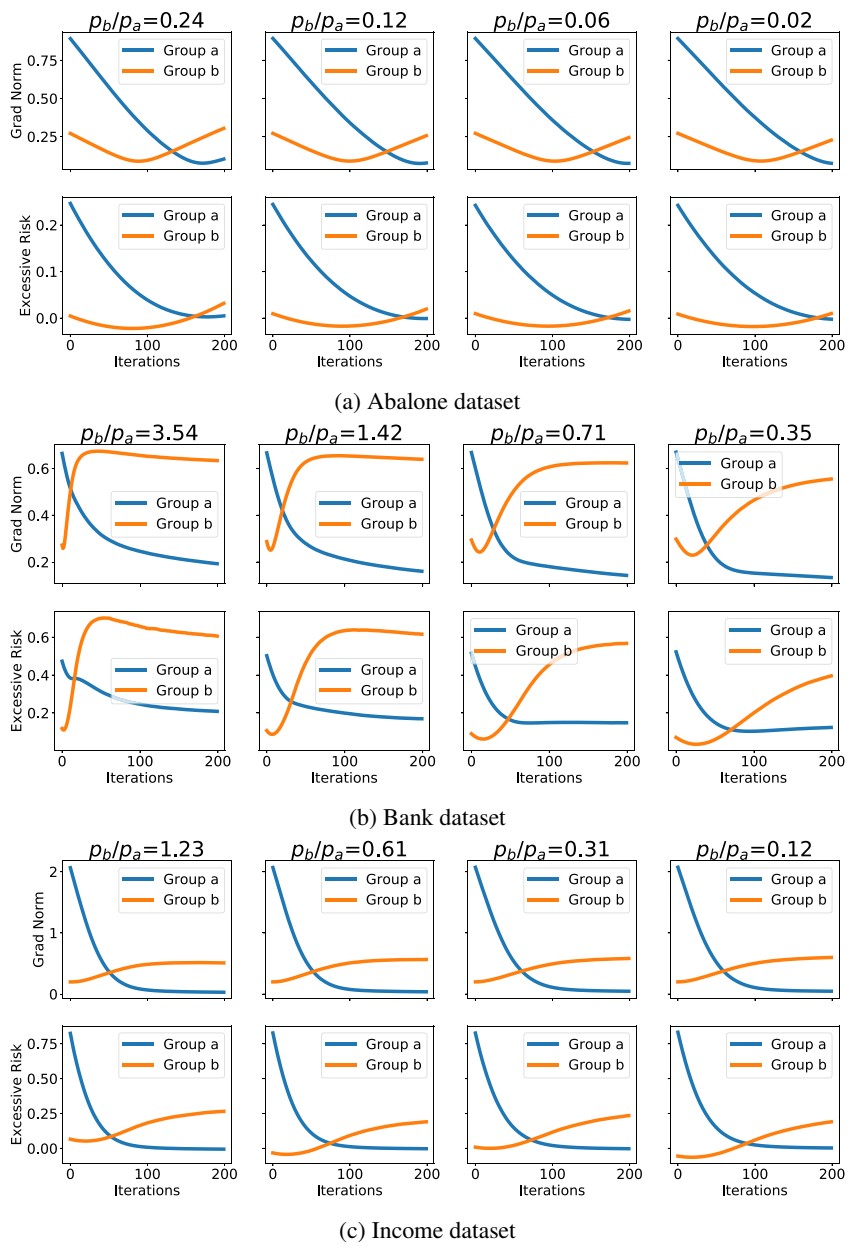

(a) Abalone dataset

(b) Bank dataset

(c) Income dataset

Figure 11: Impact of the relative group data size towards unfairness under DP-SGD (with $C = 0.1, \sigma = 5.0$).

## C.4 More on mitigation solutions

Next, this section demonstrates the benefits of the proposed mitigation solution on additional datasets. Figure 12 illustrates the excessive risk for each group in the reported datasets (recall that better fairness is achieved when the excessive risk curves values are small and similar) at varying of the privacy parameter $\epsilon$ (i.e., the excessive risk is tracked during private training).

The leftmost column in each sub-figure present the results for the baseline model, which runs DP-SGD without the proposed fairness-mitigating constraints. Observe the positive effects in reducing the inequality between the excessive risks between the groups when the solution activates both $\gamma_1$ (which regulates the component associated with $R^{clip}$) and $\gamma_2$ (which regulates the component associated with $R^{noise}$). In the reported experiments hyper-parameters $\gamma_1 = 1, \gamma_2 = 1$ were found to be good values for all our benchmark datasets. Smaller $\gamma_1$ and $\gamma_2$ values may not reduce unfairness. Likewise, large

values could even exacerbate unfairness. Using the above setting, the proposed mitigation solution was able not only to reduce unfairness in 6 out 8 cases studied, but also to increase the utility of the private models.

Once again, we mention that the design of optimal hyper-parameters is an interesting open challenge.

# D  Additional examples

## D.1  More on gradient and Hessian loss of neural networks

This section focuses on two tasks: The first is to demonstrate the connection between the gradient norm $\|g_X\|$ for some input $X$ with its input norm $\|X\|$. The second is to demonstrate the relation between the trace of the Hessian loss at a sample $X$ with input norm $\|X\|$ and the closeness of $X$ to the decision boundary. We do so by providing a derivation of the gradients and the Hessian trace of a neural networks with one hidden layer.

**Settings**  Consider a neural network model $f_{\boldsymbol{\theta}}(X) \stackrel{\text{def}}{=} softmax\left(\boldsymbol{\theta}_1^T \sigma(\boldsymbol{\theta}_2^T X)\right)$ where $X \in \mathbb{R}^d, \boldsymbol{\theta}_2 \in \mathbb{R}^{d \times H}, \boldsymbol{\theta}_1 \in \mathbb{R}^{H \times K}$ and the cross-entropy loss $\ell(f_{\boldsymbol{\theta}}(X), Y) = -\sum_{k=1}^K Y_k \log \mathbf{f}_{\boldsymbol{\theta}, \mathbf{k}}(\mathbf{X})$ where $K$ is the number of classes, and $\sigma(\cdot)$ is the a proper activation function, e.g. a sigmoid function. Let $O = \sigma(\boldsymbol{\theta}_2^T X) \in \mathbb{R}^H$ be the vector $(O_1, \dots, O_H)$ of $H$ hidden nodes of the network. Denote with $h_j = \sum_i \theta_{ji} X_i$ as the $j$-th hidden unit before the activation function. Next, denote $\boldsymbol{\theta}_{1,j,k} \in \mathbb{R}$ as the weight parameter that connects the $j$-th hidden unit $h_j$ with the $k$-th output unit $f_k$ and $\boldsymbol{\theta}_{2,i,j} \in \mathbb{R}$ as the weight parameter that connects the $i$-th input $X_i$ unit with the $j$-th hidden unit $h_j$.

**Gradients Norm**  First notice that we can decompose the gradients norm of this neural network into two layers as follows:

$$\|\nabla_{\boldsymbol{\theta}}\ell(f_{\boldsymbol{\theta}}(X), Y)\|^2 = \|\nabla_{\boldsymbol{\theta}_1}\ell(f_{\boldsymbol{\theta}}(X), Y)\|^2 + \|\nabla_{\boldsymbol{\theta}_2}\ell(f_{\boldsymbol{\theta}}(X), Y)\|^2. \tag{33}$$

We will show that $\nabla_{\boldsymbol{\theta}_2}\ell(f_{\boldsymbol{\theta}}(X), Y)\| \propto \|X\|$.

Notice that:

$$\|\nabla_{\boldsymbol{\theta}_2}\ell(f_{\boldsymbol{\theta}}(X), Y)\|^2 = \sum_{i,j} \|\nabla_{\boldsymbol{\theta}_{2,i,j}}\ell(f_{\boldsymbol{\theta}}(X), Y)\|^2.$$

Applying, Equation (14) from Sadowski [24], it follows that:

$$\nabla_{\boldsymbol{\theta}_{2,i,j}}\ell(f_{\boldsymbol{\theta}}(X), Y) = \sum_{k=1}^K (Y_k - \mathbf{f}_{\boldsymbol{\theta},k}(X))\, \theta_{1,j,k}\left(O_j(1 - O_j)\right) X_i, \tag{34}$$

which highlights the dependency of the gradient norm $\|\nabla_{\boldsymbol{\theta}_2}\ell(f_{\boldsymbol{\theta}}(X), Y)\|$ and the input norm $\|X\|^2$.

**Hessian trace**  For the connections between the Hessian trace of the loss function at a sample $X$ with the closeness of $X$ to the decision boundary and the input norm $\|X\|$, the analysis follows the derivation provided by Bishop [5]. First, notice that:

$$\text{Tr}(\boldsymbol{H}_\ell^X) = \text{Tr}(\nabla_{\boldsymbol{\theta}_1}^2 \ell(f_{\boldsymbol{\theta}}(X), Y)) + \text{Tr}(\nabla_{\boldsymbol{\theta}_2}^2 \ell(f_{\boldsymbol{\theta}}(X), Y)) \tag{35}$$

The following shows that:

1. $\text{Tr}\left(\nabla_{\boldsymbol{\theta}_2}^2 \ell(f_{\boldsymbol{\theta}}(X), Y)\right) \propto \left(1 - \sum_{k=1}^K \mathbf{f}_{\boldsymbol{\theta},k}^2(X)\right)$

2. $\text{Tr}\left(\nabla_{\boldsymbol{\theta}_1}^2 \ell(f_{\boldsymbol{\theta}}(X), Y)\right) \propto \|X\|^2$.

The former follows from Equation (26) of Bishop [5], since:

$$\nabla_{\boldsymbol{\theta}_{1,j,k}}^2 \ell(f_{\boldsymbol{\theta}}(X), Y)) = f_k(1 - f_k)O_j^2, \tag{36}$$

and thus,

$$\text{Tr}(\nabla_{\boldsymbol{\theta}_1}^2 \ell(f_{\boldsymbol{\theta}}(X), Y)) = \sum_{j=1}^H \sum_{k=1}^K f_k(1 - f_k)O_j^2 = \sum_{j=1}^H \left(\sum_{k=1}^K f_k - \sum_{k=1}^K f_k^2\right)O_j^2 = \left(1 - \sum_{k=1}^K f_k^2\right)\sum_{j=1}^H O_j^2.$$

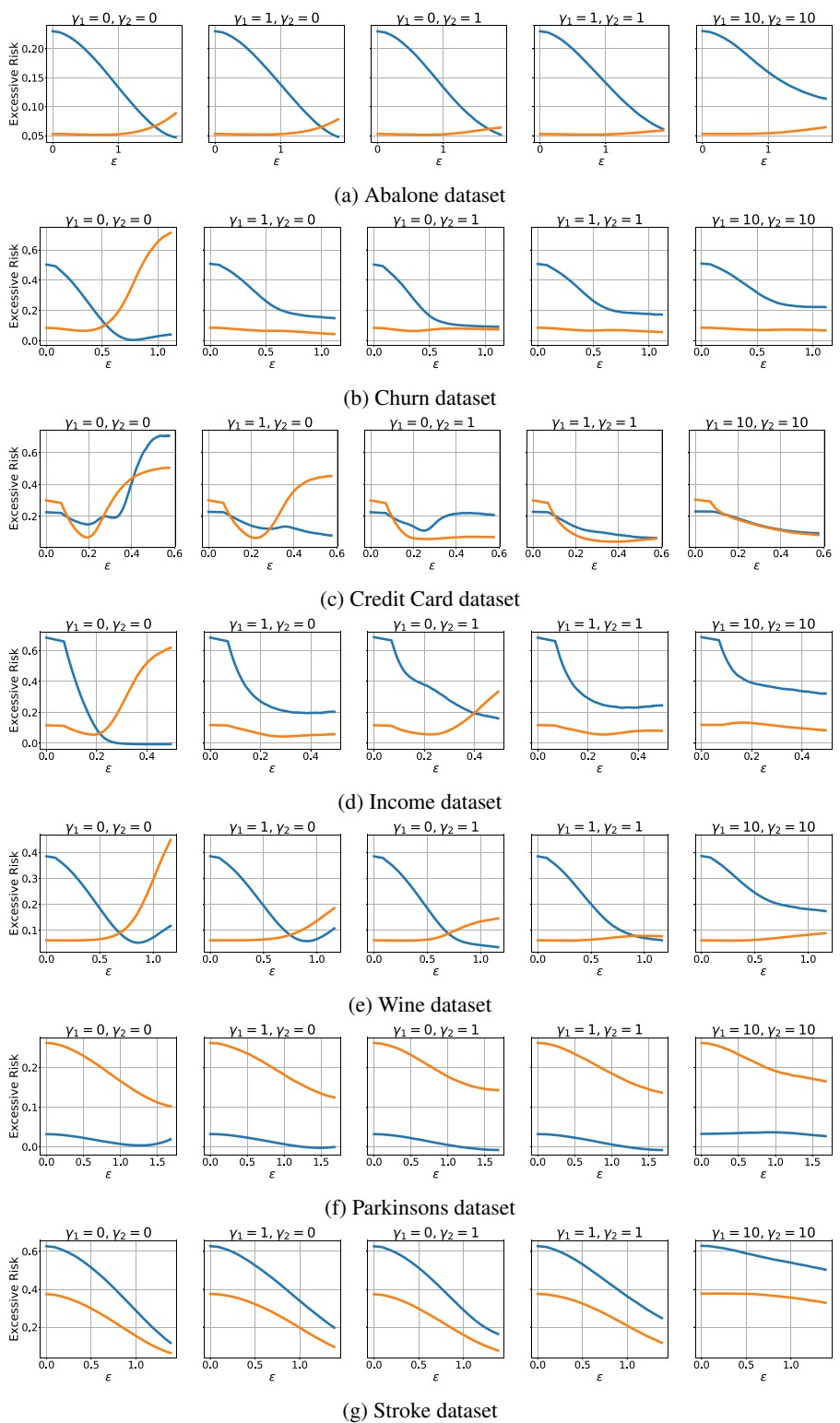

Figure 12: Mitigating solution: Excessive risk at varying of the privacy loss $\epsilon$ for different $\gamma_1$ and $\gamma_2$.

The above shows the connection between the trace of Hessian loss at a sample $X$ for the second layer of the neural network and the quantity $1 - \sum_{k=1}^{K} f_k^2(X)$ which measures how close is the sample $X$ to the decision boundary. This result relates with Theorem 5.

Regarding point (2), by applying Equation (27) of [5] we obtain:

$$\nabla^2_{\theta_{2,i,j}} \ell(f_\theta(X), Y)) = X_i^2 \Gamma_j, \tag{37}$$

where $\Gamma_j = \sigma''(h_j) \sum_{k=1}^K \theta_{2,j,k}(Y_k - f_k) + \sigma'(h_j)^2 \sum_{k=1}^K \theta^2_{2,j,k} f_k(1 - f_k)$, where $\sigma'$ and $\sigma''$ are, respectively, the first and second derivative of the activation $\sigma$ with respect to the hidden node $h_j$.

Thus:

$$\text{Tr}(\nabla^2_{\theta_2} \ell(f_\theta(X), Y)) = \sum_{j=1}^H \sum_{i=1}^d \nabla^2_{\theta_{2,i,j}} \ell(f_\theta(X), Y) = \sum_{j=1}^H \left( \sum_{i=1}^d X_i^2 \right) \Gamma_j \propto \|X\|^2,$$

which shows the dependency of the trace of the Hessian of the loss function in the first layer at sample $X$ and the data input norm.