# OpenReview forum: "Differentially Private Empirical Risk Minimization under the Fairness Lens"
_NeurIPS.cc/2021/Conference — NeurIPS 2021 Poster_

### Official Review · Reviewer_K3eh · 2021-07-04

**Rating:** 6
**Confidence:** 3

**Summary:**

The paper tries to solve the problem that ML models with DP tend to excerbate bias and unfairness. Particularly, it focuses on two well-studied DP algorithms: output perturbation and DP-SGD. The paper shows that the unfairness is caused by gradient clipping and noise addition. Based on the observations,  it further proposes a solution and its effectiveness has been tested experimentally.

**Limitations And Societal Impact:**

1 Why does Theorem 2 only show a second-order Taylor expansion of the excessive risk for group a, rather than similar result showing in Theorem 1? Since the unfairness defined in line 107 is based on excessive risk gap $\xi_a$, it is more meaningful and consistant to see the theoretical results with respect to $\xi_a$ for DP-SGD in section 6.

2  In Section 9, the paper proposes a mitigation solution with extra terms. So how to determin appropriate values for $\gamma_1$ and $\gamma_2$? How do different values of $\gamma_1$ and $\gamma_2$ affect the performance of original tasks, such as the MSE or prediction accuracy?

3 Can authors explain more about the definition of  excessive risk in line 103 and how to calculate in practice, in terms of expectation?  Since the optimal solution $\theta^*$ is not the optimal solution for the loss function w.r.t. data of group a. It can be negative values, right? But I see all excessive risk values in Figure 3 and Figure 7 are positive.  What's more, are values of excessive risk comparable among different groups? If not,  can authors explain why excessive risk is a good representation for fairness?

**Main Review:**

Originality:  The observations that both gradient clipping and noise addition cause unfairness, and the proposed mitigation solution are new and the related works are adequately cited.

Quality:  Claims are well supported by theoretical analysis and the the analyses are used appropriate. The current work includes two well-known DP algorithms and their theoretical results, demonstrations are included; Hence it is complete.

Clarity: The submission is clearly written and well organized.

Significance: The developed notion of fairness is important and interesting and other researches are likely to use the concept to build the fairness of their methods in the future.  The work narrows the gap between the unfairness of ML models and differential privacy.


**Time Spent Reviewing:**

N/A

---

> ### Author Response · Authors · 2021-08-09
> **Response to Reviewer K3eh**
>
> Thank you for the constructive feedback. We are glad the paper was well received and agree that our work may pave the way to build new fairness methods under privacy settings. We answer your questions below and will be happy to answer any further questions or doubts you may have.
>
> **Question 1: Theorem 2.**
>
> Theorem 2 decomposes the excessive risk for a given group into two components: the loss due to clipping ($R_a^{\text{clip}}$) and the loss due to noise addition ($R_a^{\text{noise}}$). The choice of using the `excessive risk` in place of the `excessive risk gap` was made to avoid superfluous notation derived by including the population level loss required by the excessive risk gap.
>
> Indeed, we are interested in highlighting the underlying factors affecting the loss due to clipping and those affecting the loss due to noise addition, as described in L199-L209.
>
> While it is simple to extend the result using the excessive risk gap, we feel that it would impact readability, due to the extra terms derived from the population level loss.
>
>
> **Question 2: Selection of $\gamma_1$ and $\gamma_2$ values.**
>
> An analysis of the effect of the parameters $\gamma_1$ and $\gamma_2$, at varying of the privacy loss $\epsilon$, was reported in Figure 14, in the Appendix. Notice how small $\gamma_1$ and $\gamma_2$ values may weakly reduce unfairness and how large values could even exacerbate unfairness.
>
> In our experiments (for all datasets and benchmarks) we set $\gamma_1 = 1$ and $\gamma_2 = 1$ (see L749-L751).
> While beyond the scope of this work, we suggested (L361-L363) that the adoption of a Lagrangian Dual framework (e.g., as in [Ref. 18]) could be a useful tool to their automatic value selection, albeit at an extra privacy cost. We think this is an interesting direction for future work.
>
> **Question 3: On the Excessive Risk.**
>
> For a group `a`, the excessive risk $R_a(\theta, D_a)$ is the difference between the expected private loss ${\cal L}(\tilde{\theta}; D_a)$ of group `a` (when the model is trained privately using the whole dataset $D$) and its non-private counterpart ${\cal L}(\theta^*; D_a$). It quantifies the accuracy drop produced by the private training. The excessive risk was approximated by sampling over 100 repetitions (see Footnote 1 on page 4). In more detail, we trained 100 private models using different random seeds and then compute the difference between the average private loss and the non-private loss values.
>
> You are correct, excessive risk can be negative (for example, in low privacy regimes) and the optimal parameters $\theta^*$ learned using the population data $D$ is often different from the optimal model parameters learned by using some group data e.g., $D_a$.
>
> Finally, the excessive risk values are _comparable_ among groups. We believe that using excessive risk to measure disparate impact under private training is a natural choice as it represents the drop in accuracy due to privacy, and we are interested in studying the disproportionate effects that privacy has on the model accuracy.
>
> We hope we have resolved all your concerns and will be happy to answer further questions you may have.

---

### Official Review · Reviewer_SVBW · 2021-07-16

**Rating:** 7
**Confidence:** 4

**Summary:**

The goal of the paper is to study the disparate impact that differential privacy can have across different groups. The authors do so within an ERM framework, and aim to equalize (or minimize the difference in) the excessive risk across groups.

**Limitations And Societal Impact:**

The authors acknowledge the limitations of their paper. In terms of societal impact, this paper is directly relevant as it aims to understand the social and disparate impact of differential privacy across groups.

**Main Review:**

Strengths:
- the excessive risk gap is a natural fairness metric here: it measures how far each group is from its optimal outcome, and compares this excess error/risk across groups. Low excessive risk gap means that all groups are *additively* as close to the best model as each other.
- It is nice that the results of the paper go beyond the classical understanding that different group sizes lead to different amount of noise added to the data, hence leading to disparate impact.
- Some of the insights in the paper are interesting: for example, the authors were able to “show” that the unfairness level is proportional to the amount of noise added for privacy
- The decomposition of Theorem 2 is really nice. It explicitly points out which part of the loss comes from the non-private loss, from clipping the gradients, and from the noise added for privacy.
- The experiments of Figure 3 shed some interesting insights as to how the choice of gradient clipping affects fairness/excessive risk.

Weaknesses:
i) The major weakness of the paper is that the math is hand-wavy and not rigorous. The results of the authors are approximations, and are not complemented with confidence intervals or some understanding of the order of magnitude of the higher order terms that are being ignored. In turn, this weakens the theoretical insights of the paper.
ii) More minor, but some of the insights of the paper are a bit hard to understand:
- Page 4 mentions the effect of the magnitude of Hessian traces on the excessive risk, but what does having a large Hessian trace means with respect to the data and the loss function? There are a few examples in the paper, but I am still having a hard time parsing what this means.
- Theorem 3 gives a sufficient condition on the magnitude of the average non-private gradient norm of the groups and the clipping value to cause unfairness, but the average gradients is something that is hard to understand theoretically, making this condition hard to use.

Question:
What happens when using different clipping values for different groups? Is it possible to intervene there to guarantee better fairness at the cost of possibly lower clipping values in some groups. This would also lead to lower sensitivity hence amount of noise to add to the gradients?


**Time Spent Reviewing:**

3-4

---

> ### Author Response · Authors · 2021-08-09
> **Response to Reviewer SVBW**
>
> Thank you for the constructive feedback. We are glad the paper was well-received, especially in light of the proposed fairness definition and analysis. We, next, address your comments and questions and will be happy to discuss and clarify any further doubt.
>
> **Approximations.**
>
> We first would like to recall that this is the first study to present insights on the underlying causes of the disparate impact produced by differentially private machine learning methods. The study has highlighted several important factors affecting the excessive risk for a given group of individuals, including the input values, the Hessian loss, the clipping terms, the privacy parameters, and the closeness to the decision boundary.
>
> The adoption of confidence intervals in sophisticated algorithms like DP-SGD goes well beyond the scope of the reported analysis, but we agree it is an important direction and it will be recognized, in the revision, as an avenue for future work. Thank you for the suggestion!
>
> **Hessian traces - intuitive explanation.**
>
> The Hessian of a function relates to the local curvature of that function at some point. In the context of the paper, the Hessian trace represents the local curvature of the loss function of points pertaining to a group, and large Hessian traces correspond to large local curvatures.
>
> Consider a group whose data samples are close to the decision boundary.  A small perturbation to the model parameters (e.g., that caused by a private training mechanism) affects the accuracy of these samples by amounts larger than those of samples lying far from the decision boundary. This leads to higher excessive risks for the former set of samples and the associated group will have larger Hessian trace values (see Theorem 5).
>
> **Theorem 3 - intuitive explanation.**
>
> Theorem 3 tells us that during private training, due to gradient clipping, groups having large average gradient norms tend to lose more information about their gradients with respect to groups with small average gradient norms.
>
> Note that our work also relates gradient norms to input norms (See paragraph "Gradient Norm" in L769 of the Appendix). Thus, practically, this result also tells us that if a group induces large gradient norms through their input features (e.g., because their data represents individuals that have, for example, much larger incomes) then this group may have higher excessive risk when compared to another group with small input norms.
>
> **Different Clipping values.**
>
> This is a very good observation. There are two factors to consider: the first is that parameterizing the clipping value by group might incur additional privacy loss to the resulting algorithm. In turn, this may negatively affect the model performance. The second is that while this option may attenuate bias due to gradient clipping, it may not be beneficial in reducing bias due to the additive noise. We will add a discussion and also notice that a study on different gradient clipping values is reported in Ref. [20].
>
> We hope we have resolved all your concerns and will be happy to answer further questions or any doubt you may have.

---

> > ### Comment · Reviewer_SVBW · 2021-08-25
> > **Thanks a lot for the response!**
> >
> > I apologize for the delay in replying to your feedback.
> >
> > With respect to the approximation, I understand the point of view of the authors. I agree that it makes sense to leave the questions of confidence intervals to future work. It would be nice, however, if possible, to have little o statements showing the terms that are not considered are much smaller than the main terms considered in the analysis of the paper, if possible?
> >
> > Wrt to the Hessian trace, I would like to thank the authors for the additional explanation and intuition. One question that I have here is: how "tight" is this closeness to the boundary interpretation? As in, are there other natural situations in which large differences in curvatures occur across groups/does this model other kinds of disparities across groups, or is the closeness to the boundary interpretation mostly without loss of generality?
> >
> > Wrt to the gradient norms, thank you for the explanation and the pointer to the Appendix. I think this pretty much answers my question here!
> >
> > Wrt to the clipping value, thank you for thinking further about this and for adding some discussion of this in the paper. This is much appreciated!

---

> > > ### Author Response · Authors · 2021-08-25
> > > **Re: Thanks a lot for the response!**
> > >
> > > Thank you for your comments!
> > >
> > > **Little o statement**: We agree and we will follow the reviewer's suggestion in the final version of our paper.
> > >
> > > **Wrt to the Hessian trace**: This is an interesting question. We note that in Section D.1 Appendix (Hessian Trace discussion) we show that the trace of the Hessian loss can be decomposed into two components: One (called here A) associated with the input norms and the other (called here B) associated with the distance to the decision boundary.
> > >
> > > While it may be difficult to determine the tightness of interpretation B, our empirical observations (not reported in the paper) have shown that in the early training stages component A dominates component B, while at convergence the converse relation occurs. This is intuitive as in the early training stages the model has not yet learned a good class separation.
> > > We thank the reviewer for posing this question. We believe that this aspect is understudied and that it outlines an interesting opportunity for future work.
> > >
> > > Thank you again for the insightful comments!

---

### Official Review · Reviewer_yKaM · 2021-07-16

**Rating:** 7
**Confidence:** 4

**Summary:**

This paper investigates the underlying causes of the disparate impact produced by differentially private machine learning methods. To that end, the paper considers the notion of excessive risk gap, which measures the difference between the population-level and subgroup risks. The paper explores how two different differential privacy mechanisms (output perturbation and DP-SGD) affect the excessive risk gap. For output perturbation, the paper relates the excessive risk gap to the absolute difference between the population-level and subgroup trace of the hessian. For DP-SGD, the paper decomposes the expected loss into a non-private term, a private term due to clipping, and a private term due to noise. Finally, the paper leverages these insights to propose a novel formulation of DP-SGD that empirically decreases the excessive risk gap on real-world datasets.

**Ethical Concerns:**

There are no ethical issues with this paper.

**Limitations And Societal Impact:**

The paper fails to discuss potential negative societal impacts within the page limit (as required by the [NeurIPS blog](https://neuripsconf.medium.com/introducing-the-neurips-2021-paper-checklist-3220d6df500b)).

**Main Review:**

**Originality:** This paper sheds light on the causes of the disparate impact produced by differentially private machine learning methods. This disparate impact was recently discovered by prior work and has not yet been fully understood. The paper presents novel insights via careful theoretical analysis of the effects of differential privacy methods on excessive risk.

**Quality:** The submission is technically sound, and the proofs are generally correct. However, there are some (mostly minor) mistakes:
- The term $\nabla_{\theta^*} \mathcal{L}(\theta^*, D_a)$ is missing in the expectation of L506.
- Equations 15b, 15c, 15d, and 16 all mistakenly use $\bar{g}_B$ instead of $\bar{g}_D$.
- You probably also need to assume that $\eta \leq 1$ since $\eta$ is squared in Equation 25d.
- Equation 28c is strictly greater than the right-hand side of Equation 29a.

**Clarity:** The paper is clearly written and well organized. However, certain presentational aspects could be improved (also see questions below). The most critical ambiguity is the paper’s claim on L134 that `groups with larger Hessian traces will have larger excessive risk than groups with smaller Hessian traces`. It is unclear why this statement would make sense since the excessive risk is approximated by the absolute difference of the population-level and subgroup hessian traces. Thus the magnitude of the subgroup Hessian trace is entirely irrelevant if it is close to the population-level Hessian trace. Furthermore, the adjacency relation and range should be defined for the Gaussian mechanism on L083. Finally, there are a couple of errors:
- DP-SGD is frequently misspelled as DP-SDG
- L025 lending instead of landing
- L125 from instead of from
- L140 varying *levels* of
- L322 these instead of this
- L604 it instead of if

**Significance:** The submission advances the understanding of the interplay between differential privacy and fairness in machine learning. The results could be useful to researchers from both fields when designing novel fairness of privacy approaches.

**Questions**

- Section 10 mentions that the proposed mitigation solution negatively affects the training runtime but does not provide any concrete numbers. How much slower is the proposed solution?
- How can the excessive risk gap values in Figure 8 be negative if the excessive risk gap is the absolute difference between two risk functions?
- What happened to the term $- p_a p_b g_{D_a} g_{D_b}$ when going from Equation 27b to Equation 28a?
- Is the global sensitivity in L081 and L118 the same?
- Why is the convexity assumption necessary in Theorem 1? Is this only required for output perturbation?
- Why does the privacy analysis of output perturbation require optimality of the model parameters? Could one not apply output perturbation to non-optimal parameters?

I am happy to increase my score if the authors can resolve my concerns.

**Time Spent Reviewing:**

8.5

---

> ### Author Response · Authors · 2021-08-09
> **Response to Reviewer yKaM**
>
> Thank you for the constructive feedback. We are glad the paper was well received. We address your comments as follows and will be happy to discuss and clarify any further doubts.
>
> **Presentation and minor typos.**
>
> We sincerely appreciate your time and review. We will correct all typos pointed out in your revision.
>
> **Ambiguity on L134.**
>
> Firstly, let us recall that the `excessive risk` is defined as the difference between the expected private risk function and its non-private counterpart (Equation (1)). The approximation (Equation (3)) mentioned in the comment refers to the `excessive risk gap`, which is defined as the difference between the excessive risk of the population and the group levels (Equation (2)).
> We also recall that the excessive risk for a group `a` can be approximated as $\text{const} \times \text{Tr}(H_\ell^a)$ (see L127). From it, it follows that groups with larger Hessian losses may have larger excessive risks. We do acknowledge the two terms are similar and we will emphasize this aspect to make it clearer.
>
> ### Questions
>
> - **Runtime:** Using the settings and datasets described in the paper, the proposed solution can be 2 to 8 times slower than the (non-fair) baseline. This is due to the computations of the average clipped gradient $\bar{g}_D$ (see the Equation after L344).
> We suspect that the use of modern frameworks like JAX may mitigate this slowdown --- JAX was shown to speed up the computations of DP-SGD up to 50x [$\dagger$]. However, since the paper did not focus on runtime, we have omitted these results. We'll be happy to include them in the final version if the reviewer finds them useful. In fact, this point may recognize an avenue for future work.
>
> - **Figure 8:** We realize that the y-axis was mistakenly labeled as _excessive risk gap_. Instead, it represents the _excessive risk_ of each group and the figure shows the correlation between the excessive risk and the Hessian traces. The same miss-label appears in Figure 1. The typo will be corrected in the final version. Thank you very much for this pointing this out!
>
> - **Equations 27b-28a:** The absence of this term in the proof was a frustrating omission when going from Equations (27b) to (28a). Thank you for pointing it out! It will be corrected. Note that this does not affect any of the reported experiments or conclusions, albeit the LHS of Equation (5) becomes $\\| g_{D_a}\\| (\\frac{p_a^2}{2})$. We sincerely appreciated your careful revision.
>
> - **Global Sensitivity:** Yes, the global sensitivities in L081 and L118 are the same.
>
> - **Convexity Assumption:** The convexity assumption is necessary for output perturbation and is used to derive the global sensitivity in L118 (see Ref [7] of the main paper). DP-SGD, however, does not require convexity of the loss function. Our Theorem 2 does not require a convexity assumption.
>
> - **Optimality of Output Perturbation:** The optimality of output perturbation is required to provide its privacy guarantees (see Algorithm 1 and Theorem 6 of Ref [7]) as well as its performance guarantees (See Theorem 15 and Lemma 16 in Ref [7]).
>
> [$\dagger$] _Pranav Subramani, Nicholas Vadivelu, Gautam Kamath. "Enabling Fast Differentially Private SGD via Just-in-Time Compilation and Vectorization". Corr/Abs: 2010.09063, 2020._
>
> We hope to have resolved all your concerns and will be happy to answer further questions you may have. Thank you again for your comments and constructive feedback.

---

> > ### Comment · Reviewer_yKaM · 2021-08-23
> > **Thank You for Resolving My Concerns**
> >
> > I thank the authors for their response, which addresses my questions and concerns. I would definitely include the running time results in the final version.

---

> > > ### Author Response · Authors · 2021-08-24
> > > **Re: Thank You for Resolving My Concerns**
> > >
> > > We are glad we could resolve all your concerns.
> > > Thank you also for your suggestion. We will include the runtime results in the final version, in addition to the revisions pointed out in our responses.
> > >
> > > Many thanks again!

---

### Decision · Program_Chairs · 2021-09-27

**Decision:**

Accept (Poster)

**Comment:**

The paper studies impact of various DP algorithms on fairness. The paper is well written and easy to follow.  I encourage authors to incorporate reviewer suggestions and make the analysis more rigorous in the final version of the paper e.g., change Theorem 1 to include higher order error terms via O() notation.